# BI-STRIDE MULTI-SCALE GRAPH NEURAL NETWORK FOR MESH-BASED PHYSICAL SIMULATION

## ABSTRACT

Learning physical systems on unstructured meshes by flat Graph neural networks (GNNs) faces the challenge of modeling the long-range interactions due to the scaling complexity w.r.t. the number of nodes, limiting the generalization under mesh refinement. On regular grids, the convolutional neural networks (CNNs) with a U-net structure can resolve this challenge by efficient stride, pooling, and upsampling operations. Nonetheless, these tools are much less developed for graph neural networks (GNNs), especially when GNNs are employed for learning large-scale mesh-based physics. The challenges arise from the highly irregular meshes and the lack of effective ways to construct the multi-level structure without losing connectivity. Inspired by the bipartite graph determination algorithm, we introduce Bi-Stride Multi-Scale Graph Neural Network (BSMS-GNN) by proposing *bi-stride* as a simple pooling strategy for building the multi-level GNN. *Bi-stride* pools nodes by striding every other Breadth-First-Search (BFS) frontier; it 1) works robustly on any challenging mesh in the wild, 2) avoids using a mesh generator at coarser levels, 3) avoids the spatial proximity for building coarser levels, and 4) uses non-parametrized aggregating/returning instead of MLPs during pooling and unpooling. Experiments show that our framework significantly outperforms the state-of-the-art method's computational efficiency in representative physics-based simulation cases.

## 1 INTRODUCTION

Simulating physical systems through numerically solving partial differential equations (PDEs) plays a key role in various science and engineering applications, ranging from particle-based (Jiang et al., 2016) and mesh-based (Li et al., 2020a) solid mechanics to grid-based fluid (Bridson, 2015) and aero (Cao et al., 2022) dynamics. Despite the extensive successes in improving their stability, accuracy, and efficiency, numerical solvers are often computationally expensive for time-sensitive applications, especially iterative design optimization where fast online inferring is desired.

Recently, machine learning approaches have demonstrated impressive potential in improving the efficiency of inferring physical states with competitive accuracy. Representative methods include end-to-end frameworks (Obiols-Sales et al., 2020) and those with physics-informed neural networks (PINNs) (Raissi et al., 2019; Karniadakis et al., 2021; Sun et al., 2020; Gao et al., 2021). Many existing works apply convolutional neural networks (CNNs) (Fukushima & Miyake, 1982) to learn physical systems on two- or three-dimensional structured grids (Kim et al., 2019; Fotiadis et al., 2020; Gao et al., 2021; Guo et al., 2016; Tompson et al., 2017). It is generally recognized that CNNs exhibit strong performance on handling local information with convolution and global information with pooling/upsampling. However, the strict dependency on regular domain shapes makes it non-trivial to be applied on unstructured meshes. Although it is possible to deform the domains to rectangular shapes to apply CNNs (Gao et al., 2021) or other models, such as NeuralOperatorNets (Li et al., 2022), the challenge remains for domains with complex topologies, which are common in practice.

On the other hand, graph neural networks (GNNs) have been considered as a natural choice for physics-based simulation on unstructured meshes (Battaglia et al., 2018; Belbute-Peres et al., 2020; Gao et al., 2022; Harsch & Riedelbauch, 2021; Pfaff et al., 2020; Sanchez-Gonzalez et al., 2018; 2020). However, all the above methods use the flat GNN that faces two challenges when the graph

size increases: (1) **Oversmoothing**: the graph convolution can be seen as a low-pass filter that suppresses the signal with higher frequency than a certain value (Chen et al., 2020; Li et al., 2020b). Multiple passes of graph convolution then become an iterative projection onto the eigenspace of the graph where all higher frequency signals are smoothed out, which also makes training harder. (2) **Complexity**: Under mesh refinement, not only that more nodes are there to be processed, but the message passing (MP) iterations also grow linearly to propagate information to the same physical distance (Fortunato et al., 2022). As a result, a quadratic complexity becomes inevitable for both the running time and the memory to store the computational graph.

To mitigate these limitations, researchers recently start investigating multi-scale GNNs (MS-GNNs) for physics-based simulation (Fortunato et al., 2022; Li et al., 2020b; Lino et al., 2021; Liu et al., 2021; Lino et al., 2022a;b). The multi-scale approach is appealing as it tackles the **oversmoothing** issue by building sub-level graphs on coarser resolutions, which lead to longer range interaction and naturally fewer MP times. However, pooling and adjacency building should be conducted carefully to avoid introducing partitions into the coarser levels (Gao & Ji, 2019), which stops information exchange across the separated clusters. Existing solutions include utilizing the spatial proximity for building the connections at the coarser levels (Lino et al., 2021; Liu et al., 2021; Lino et al., 2022a;b), or generating coarser meshes for the original geometry (Fortunato et al., 2022; Liu et al., 2021), and randomly pooling nodes then applying Nyström approximation for the original adjacency matrix (Li et al., 2020b). However, all of them suffer from limitations: the spatial proximity can result in wrong connections across the geometry boundaries; the mesh generation is laboring and often unavailable for unseen meshes; and the random pooling may introduce partitions in the coarser levels.

We observe that all the aforementioned limitations originate from *pooling* and *building connections at coarser levels*. To the best of our knowledge, no existing work can systematically generate multi-scale GNNs with arbitrary levels for an arbitrary geometry in the wild while completely avoiding cutting or wrong connections across the boundaries. To this end, in this work, we introduce a simple yet robust and effective pooling strategy, *bi-stride*. Bi-stride is inspired by the bi-partition determination in DAG (directed acyclic graph). It pools all nodes on every other BFS (breadth-first-search) frontiers, such that a $2^{nd}$-powered adjacency enhancement conserves all the connectivity. We also accompany bi-stride with a non-parameterized aggregating/returning method to handle the transition between adjacent levels to decrease the model complexity. Our framework, namely *Bi-Stride Multi-Scale Graph Neural Network* (*BSMS-GNN*), is tested on three benchmarks (CYLINDERFLOW, AIRFOIL, and DEFORMINGPLATE) from GraphMeshNets and INFLATINGFONT, a new dataset of inflating elastic surfaces with many self-contacts. In all cases, BSMS-GNN shows a dominant advantage in memory footprint and required training and inference time compared to alternatives.

## 2 BACKGROUND AND RELATED WORKS

**GNNs for Physics-Based Simulation**   GNNs are first applied to physical simulation to learn the behaviors of particle systems, deformable solids, and Lagrangian fluids (Battaglia et al., 2016; Chang et al., 2016; Mrowca et al., 2018; Sanchez-Gonzalez et al., 2020). Notably, the generalized Message Passing (Sanchez-Gonzalez et al., 2018) is broadly accepted for information propagation. Based on that, GraphMeshNets (Pfaff et al., 2020) sets a milestone for learning mesh-based simulation. Following GraphMeshNets, which predicts a single forward timestep, there have been several variants, including 1) solving forward and inverse problems by combining GNNs with PINNs (Gao et al., 2022), 2) predicting long-term system states combined with GraphAutoEncoder (GAE) and Transformer (Han et al., 2022), 3) predicting steady states with multi-layer readouts (Harsch & Riedelbauch, 2021), and 4) up-sampling from coarser meshes with differentiable simulation (Belbute-Peres et al., 2020). Yet still, with flat GNNs, the quadratic computation complexity on finer meshes poses great challenges. We claim that adopting a multi-level structure is an effective solution.

**Multi-Scale GNNs**   It is common to apply GNNs with multi-level structures in various graph-related tasks, such as graph classification (Wu et al., 2020; Mesquita et al., 2020; Zhang et al., 2019). GraphUNet (GUN) (Gao & Ji, 2019) first introduces the UNet structures into GNN with a trainable scoring module for pooling; it also has a $2^{nd}$-powered adjacency enhancement to reduce the chance of losing connectivity. A few works have investigated multi-scale GNNs (MS-GNNs) for physics-based simulation. Specifically, Fortunato et al. (2022) and Liu et al. (2021) define two- and multi-level GNNs, respectively, for physics-based simulation, but both of them rely on pre-

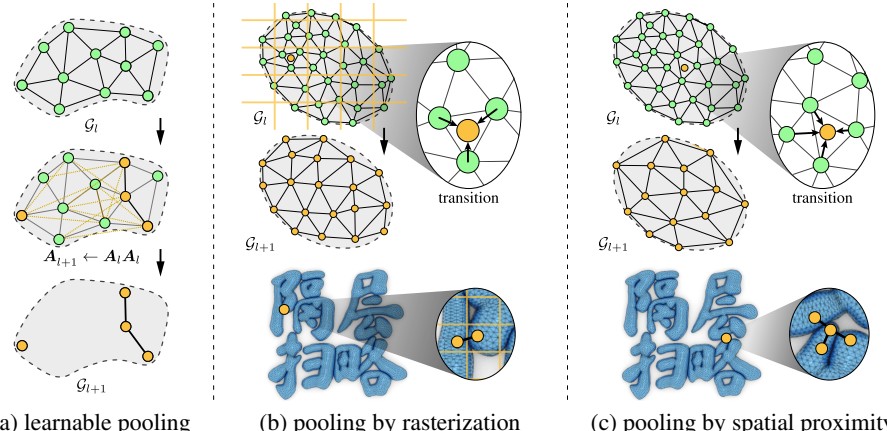

Figure 1: **Issues of existing multi-level GNNs.** (a) A learnable pooling (Gao & Ji, 2019) may lead to loss of connectivity even after $1^{\text{st}}$-order enhancement. (b) A pooling by rasterization (Lino et al., 2021; 2022a;b) and (c) by spatial proximity (Liu et al., 2021; Fortunato et al., 2022) can lead to wrong connections across the boundaries at the coarser level.

generated coarse meshes. Lino et al. (2021; 2022a;b) use the original mesh at the first level and project it to regular grids (MS-GNN-GRID) at the coarser levels. Li et al. (2020b) adopt multi-level matrix factorization to generate the kernels at arbitrary levels without requiring mesh generators or K-nearest neighbor (K-NN) interpolations. Concerning building the connections and hierarchies on point clouds with radius samplers, there are representative works such as GNS Sanchez-Gonzalez et al. (2020), PointNet Qi et al. (2017a), PointNet++ Qi et al. (2017b), and GeodesicConv Masci et al. (2015). Still, these methods by construction suit better cases without meshes, such as particle fluid simulations.

**Motivations of Our Method** We present an overview of representative GNN architectures with U-net structure in Fig. 1. Two major disadvantages we observed are: 1) easy loss of connectivity by pooling, even with a $2^{\text{nd}}$-powered adjacency enhancement; and 2) lack of direct connections between pooled and unpooled nodes, leading to additional edges built by the spatial proximity for transition between levels.

For a more clear illustration, we start with a few definitions. We first define the adjacency enhancement by the $K^{\text{th}}$-order matrix power as $\boldsymbol{A} \leftarrow \boldsymbol{A}^K$, where $\boldsymbol{A}$ is the adjacency matrix of the graph. Geometrically, $\boldsymbol{A}(i,j) = 1$ means the edge $(i,j)$ exists, and $\boldsymbol{A}^K(i,j) = 1$ means that node $j$ is connected to node $i$ via at most $K$ hops. Given a pooling strategy P and the selected pooled nodes $\mathcal{S}_{\text{P}}$, we define a $K$th-order outlier set as $\mathcal{O}_K$, where the nodes in $\mathcal{O}_K$ are not connected to any pooled nodes even after $K^{\text{th}}$-order adjacency enhancement: $\boldsymbol{A}^K(i,j) = 0, \forall i \in \mathcal{S}_{\text{P}}, \forall j \in \mathcal{O}_K$.

We further define that a pooling strategy P is $K^{\text{th}}$-order connection conservative (K-CC) if $\mathcal{O}_K$ is empty. We argue that larger $K$ in $K^{\text{th}}$-order adjacency enhancement is harmful to distinguish the node features. As $K$ increases, $\boldsymbol{A}^K(i,j)$ approaches a matrix with all its entries equal to 1, representing a fully connected graph, where a single step of convolution will average all node features and make them indistinguishable. The most favorable and possible, i.e. the smallest, $K$ we should seek is 2. Gao & Ji (2019) uses the $2^{\text{nd}}$ order enhancement to help conserve the connectivity. Nonetheless, there is no theoretical guarantee that a learnable pooling module is consistently 2-CC for any graph (a counter example is shown in Fig. 1(a)). There are two alternative solutions to the matrix power enhancement that ensure conservation of the connectivity at coarser layers: 1) Lino et al. (2022a; 2021; 2022b) build the coarser graph by projecting the finer nodes to the nearby background grids (Fig. 1(b)); 2) Liu et al. (2021); Fortunato et al. (2022) create coarser meshes for the same domain (Fig. 1(c)). However, both methods need spatial proximity to build additional connections for the transition between levels, which may produce wrong connections across the boundary. These limitations motivate us to create a consistent 2-CC pooling strategy, as described in Sec. 3.2.

An additional overhead is the learnable transition modules which have the same network architecture as the message passing. This overhead in model size and computational complexity grows linearly w.r.t. the number of levels of U-net. As a result, they often end up with a relatively shallow level at 2

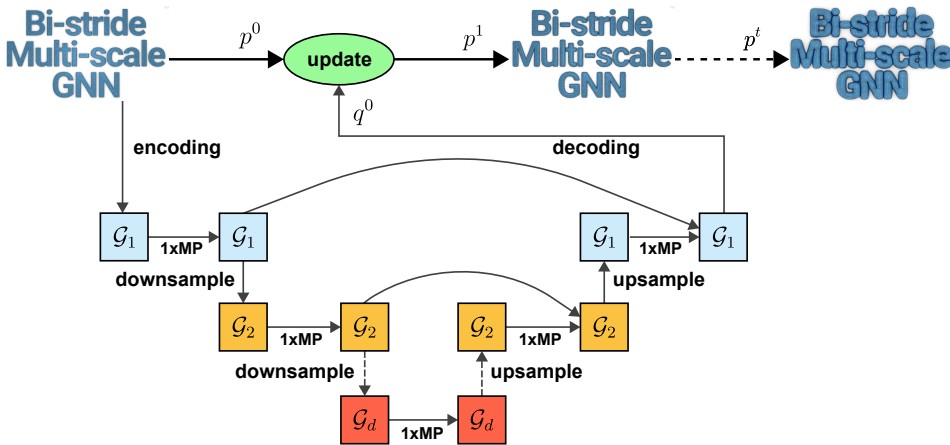

Figure 2: **BSMS-GNN pipeline** uses encode-process-decode trained with one-step supervision. $\mathcal{G}_1, \mathcal{G}_2, \cdots, \mathcal{G}_d$ represents the graph at different levels (finest to coarsest). The encoder/decoder only connects the input/output fields with the latent fields at $\mathcal{G}_1$. The latent nodal fields are updated by one MP (message passing) at each level. The bi-stride pooling selects the pooled nodes for the adjacent coarser level, and the transition is conducted in a non-parameterized way.

or 3. We claim that none-parameterized transition is performance-wise crucial for deeper multi-layer GNNs, and propose the first non-parameterized transition method in Sec. 3.3.

Overall, our method adopts a similar message passing layer as in GraphMeshNets (Pfaff et al., 2020). Compared to Liu et al. (2021); Fortunato et al. (2022), our advantage is that no mesh generators is needed for the coarser-level graphs. Compared to Lino et al. (2021; 2022a;b), our advantage is that no spatial proximity is necessary. Together, we eliminate the need for building connections via spatial proximity nor using learnable MLP for aggregation and returning. Note that the work of Li et al. (2020b) shares similar advantages to some extent, but it focuses on generalization with PDE parameters, while ours focuses on a systematic pooling strategy for arbitrary complex geometries.

## 3 METHODOLOGY

### 3.1 DEFINITIONS

Figure. 2 presents the overall structure of BSMS-GNN. We consider the evolution of a physics-based system discretized on a mesh, which is converted to an undirected graph $\mathcal{G}_1 = (\mathcal{V}_1, \mathcal{E}_1)$. Here, with subscript 1, $\mathcal{V}_1$ and $\mathcal{E}_1$ label the nodal fields and the connectivities at the finest level (the input mesh), respectively. Specifically for edges, we define $\mathcal{E}_1 = \{\mathcal{E}_1^1, \cdots, \mathcal{E}_1^S\}$, where $\mathcal{E}_1^1$ is the edge set directly copied from the input mesh, and $\{\mathcal{E}_1^k|_{k=2}^S\}$ are optionally the additional problem-dependent edge sets involved. For example, both DEFORMINGPLATE (Fig. 5(c)) and INFLATINGFONT (Fig. 5(d)) benchmarks have a second edge set $\mathcal{E}_1^2$ for the nearby colliding vertices. We use $\{p, q\}$, stacked vectors of $\{p_i, q_i\}$ of all nodes $i \in \mathcal{V}_1$, to denote the input and output nodal fields, respectively. Given an input field $p^j$ at a previous time $t_j$, one pass of our BSMS-GNN returns the output field $q^{j+1}$ at time $t_{j+1} = t_j + \Delta t$, where $\Delta t$ is the fixed time step size. The output $q$ can contain more physical fields than the input $p$ and must be able to derive the input for the next pass. The rollout refers to iteratively conducting BSMS-GNN from the initial state $p^0 \to q^1 \to p^1 \to \cdots \to q^n$ and producing the temporal sequence output $\{q^1, q^2, \cdots, q^n\}$ within the time range of $(t_0, t_0 + n\Delta t]$, where $n$ is the total number of evaluations.

**Message Passing** In general, we follow the *encode-process-decode* fashion in GraphMeshNets, where encoding and decoding only appear at the beginning and the end of the finest level $\mathcal{G}_1$, mapping the nodal input $p$ and output $q$ to/from the latent feature $\boldsymbol{v}$, respectively (see Table A.1 for the domain-specific information). As for the process part, unlike GraphMeshNets where multiple message passings (MPs) are needed, we observe that a single MP at each level is sufficient for all experiments. Therefore, it becomes unnecessary to keep updating the latent edge information across multiple MPs. To include the directional information of an edge $(\boldsymbol{x}_i, \boldsymbol{x}_j)$, we simply prepend its

positional offset $\Delta \boldsymbol{x}_{ij} = \boldsymbol{x}_i - \boldsymbol{x}_j$ to the stacked sender/receiver latent as input to calculate the information flow. For a problem involving $S$ edge sets, an MP pass at level $l$ is formulated as:

$$
\begin{aligned}
\boldsymbol{e}^s_{l,ij} &\leftarrow \mathrm{f}^s_l\big(\Delta\boldsymbol{x}_{l,ij}, \boldsymbol{v}_{l,i}, \boldsymbol{v}_{l,j}\big), \quad s = 1, \cdots, S, \\
\boldsymbol{v}'_{l,i} &\leftarrow \boldsymbol{v}_{l,i} + \mathrm{f}^V_l\Big(\boldsymbol{v}_{l,i}, \sum_j \boldsymbol{e}^1_{l,ij}, \cdots, \sum_j \boldsymbol{e}^S_{l,ij}\Big),
\end{aligned}
\tag{1}
$$

where f is a MLP function, $\boldsymbol{e}$ is the latent information flow through an edge, and $\boldsymbol{v}$ is the latent node feature. Please refer to Sec. A.2 for the detailed architecture of the model.

**Cross-Level Transition** We handle information transition between two adjacent levels with *downsampling* and *upsampling* modules. Here we define downsampling as the sequence of pooling (selecting pooled nodes) and then aggregating the information from the neighbors to the coarser level, and we define upsampling as the sequence of unpooling and then returning the information of the pooled nodes to their neighbors at the finer level. Please refer to Sec. 3.3 for details.

## 3.2  Bi-Stride Pooling and Adjacency Enhancement

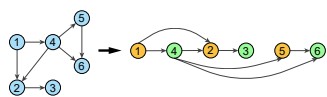

(a) Bi-partition of a DAG

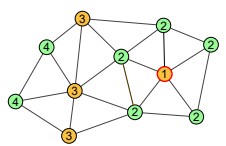

(b) Bi-stride of a mesh

As aforementioned, two challenges of building a multi-level GNN for learning physical simulation, especially on wild geometries, are 1) not introducing partitions that break the connectivity, and 2) not introducing wrong edges by spatial proximity. We tackle these challenges by improving the pooling phase. Specifically, is there a pooling strategy that is consistently 2$^\text{nd}$-order connection conservative (2-CC) for any input graph so that an efficient 2$^\text{nd}$-order enhancement is sufficient to conserve the connectivity? We draw the initial inspiration from the bi-partition determination algorithm (Asratian et al., 1998) in a directed acyclic graph (DAG). As show in the inset figure (a), after topological sorting, pooling on every other depth (yellow and green) generates a bi-partition. To resemble the bi-partition determination on a mesh, which is not bi-partite due to cycles, we can conduct a breadth-first search (BFS) to compute the geodesic distances from an initial seed to all other nodes, and then stride and pool all nodes at every other BFS frontiers (bi-stride). A bi-stride example is shown in the inset figure (b), where the number in each vertex represents the distance to the seed (node 1 in red circle) by BFS. This pooling is 2-CC by construction and conserves direct connections between pooled nodes and unpooled nodes. As a result, we avoid building edges by spatial proximity or handling the cumbersome corner cases such as cross-boundary connections.

**Seeding Heuristics** We claim that there should exist some freedom as long as the seeding is balanced to a certain degree. The time complexity for searching seeds is tolerable because of the one-pass preprocess. For training datasets, we choose two deterministic seeding heuristics: 1) closest to the center of a cluster (CloseCenter) for INFLATINGFONT, and 2) the minimum average distance (MinAve) for all other cases, and we preprocess the multi-level building in one pass. One can consider the cheaper heuristic CloseCenter during the online inferring phase if an unseen geometry is encountered. The details of the algorithms can be found in Sec.A.6.

**Auxiliary Edges** For multi-physics problems, such as DEFORMINGPLATE (Fig. 5(c)) and INFLAT-INGFONT (Fig. 5(d)), the auxiliary edges (such as contact edges $\boldsymbol{A}^C$) should be built dynamically by spatial proximity to exchange the interfacial information between different systems. The enhancement of these edges should be handled properly for multi-layered GNN, which, to the best of our knowledge, has not been addressed yet. At level $l$, given two adjacent matrices $\boldsymbol{A}_l$ and $\boldsymbol{A}_l^C$ for the mesh edges from the input graph $\mathcal{G}_l^1$ and the contact edges, respectively, we apply the enhancement followed by per-cluster bi-stride pooling for $\boldsymbol{A}_l$ with selected node indices $\mathcal{I}$:

$$
\begin{aligned}
\boldsymbol{A}'_{l+1} &\leftarrow \boldsymbol{A}_l \boldsymbol{A}_l, \quad \boldsymbol{A}_{l+1} \leftarrow \boldsymbol{A}'_{l+1}[\mathcal{I}, \mathcal{I}], \\
\boldsymbol{A}'^C_{l+1} &\leftarrow \boldsymbol{A}_l \boldsymbol{A}_l^C \boldsymbol{A}_l, \quad \boldsymbol{A}^C_{l+1} \leftarrow \boldsymbol{A}'^C_{l+1}[\mathcal{I}, \mathcal{I}].
\end{aligned}
\tag{2}
$$

This enhancement can be geometrically interpreted as such: an auxiliary edge $(i, j)$ should exist if $j$ is reachable from $i$ in 2 hops and one of which is an auxiliary edge at the finer level. We prove in Sec. A.5 that our pooling conserves all the contact edges under this enhancement.

## 3.3 TRANSITION BETWEEN LEVELS

We propose a unified non-parameterized method to reduce the overhead of the learnable transition modules between every pair of adjacent levels.

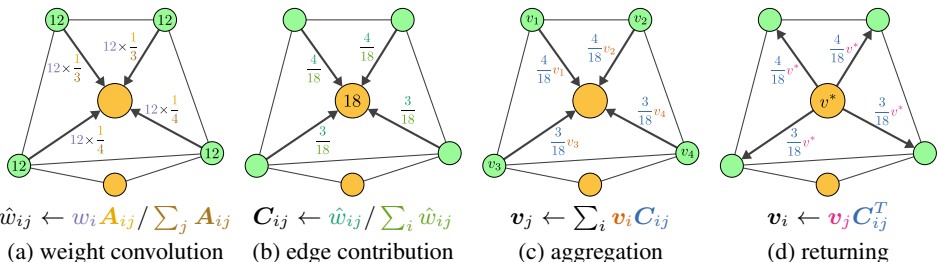

$$\hat{w}_{ij} \leftarrow w_i \boldsymbol{A}_{ij} / \textstyle\sum_j \boldsymbol{A}_{ij} \qquad \boldsymbol{C}_{ij} \leftarrow \hat{w}_{ij} / \textstyle\sum_i \hat{w}_{ij} \qquad \boldsymbol{v}_j \leftarrow \textstyle\sum_i \boldsymbol{v}_i \boldsymbol{C}_{ij} \qquad \boldsymbol{v}_i \leftarrow \boldsymbol{v}_j \boldsymbol{C}_{ij}^T$$

(a) weight convolution    (b) edge contribution    (c) aggregation    (d) returning

Figure 3: **Schematic plot of the transition steps between adjacent levels.**

**Downsampling** We treat the latent information as a conserved variable and project it to the pooled nodes. We define $\boldsymbol{A}$ as the unweighted adjacency matrix where its row and column indices represent the sender and the receiver, respectively. We further represent the nodal mass or importance as a nodal weight field $w$, which is initialized on the finest level to ones for near-uniform meshes or the volume/mass field for highly irregular meshes. With the receiver vertex $j$ and its sender vertices $i$, the formal procedure is formulated as (Fig. 3):

- Normalize by row as in a standard graph convolution $\hat{\boldsymbol{A}}_{ij} \leftarrow \boldsymbol{A}_{ij} / \sum_j \boldsymbol{A}_{ij}$, and then convolve the weight once $\hat{w}_{ij} \leftarrow w_i \hat{\boldsymbol{A}}_{ij}$ (Fig. 3(a));
- Calculate edge weights $\boldsymbol{C}_{ij} \leftarrow \hat{w}_{ij} / \sum_i \hat{w}_{ij}$, where $\boldsymbol{C}$ can be viewed as a contribution table with $\boldsymbol{C}_{ij}$ as the share of weights in the receiver $j$ contributed by the sender $i$ (Fig. 3(b));
- Convolve the latent information by the contribution table $\boldsymbol{v}_j \leftarrow \sum_i \boldsymbol{v}_i \boldsymbol{C}_{ij}$, which is equivalent to equally splitting and sending the weighted information to neighbors, and then obtaining the weighted average (Fig. 3(c)).

**Upsampling** After unpooling, all nodes except the pooled ones have zero information. A returning process, resembling the transposed convolution in CNNs, can help distinguish the receivers. With the contribution table $\boldsymbol{C}$ recording the edge weights, a natural choice is $\boldsymbol{v}_i \leftarrow \boldsymbol{v}_j \boldsymbol{C}_{ij}^T$ (Fig. 3(d)).

## 4 EXPERIMENTS

### 4.1 EXPERIMENT SETUP

**Datasets** We adopt three representative public datasets from GraphMeshNets (Pfaff et al., 2020): 1) CYLINDERFLOW: incompressible fluid around a cylinder where the mass conservation has to be enforced globally, 2) AIRFOIL: compressible flow around an airfoil where an auxiliary prediction, the pressure, is included, and 3) DEFORMINGPLATE: deforming an elastic plate with an actuator where simple contact is included. In addition, to illustrate the ease of extending our method to multiset problems, we further create a new dataset, INFLATINGFONT, featuring the inflation of enclosed elastic surfaces with massive self-contacts (Fang et al., 2021).

**Baselines** On all datasets, we compare computational complexity, training/inference time, and memory footprint of BSMS-GNN to baselines: 1) GRAPHMESHNETS (Pfaff et al., 2020): the single-level GNN architecture of GraphMeshNets, 2) MS-GNN-GRID (Lino et al., 2021; 2022a;b): a representative work for those building the hierarchy with spatial proximity (i.e. using the distance between nodes), and 3) GRAPHUNET (Gao & Ji, 2019): a representative work for those using learnable modules for pooling. The detailed reimplementation of these works can be found in Sec. A.2. We note again that methods such as Liu et al. (2021); Fortunato et al. (2022) are not practical because they require pre-drawing meshes at multiple levels. For all cases reported in this work, this means manually drawing $20,000$ meshes using CAE or meshing software.

| Measurements | Case | Our's | Lino et al. (2021) | Pfaff et al. (2020) | Gao & Ji (2019) |
|---|---|---|---|---|---|
| Training time/step [ms] | Cylinder | **10.14** | 15.36 | 19.29 | 16.20 |
| | Airfoil | **18.82** | 25.26 | 36.72 | 55.08 |
| | Plate | **15.58** | 49.65 | 49.15 | 31.88 |
| | IDP | **45.96** | 107.16 | 117.48 | 1,833.37 |
| Infer time/step [ms] | Cylinder | 6.75 | **6.18** | 14.50 | 24.30 |
| | Airfoil | **8.64** | 20.40 | 24.20 | 33.60 |
| | Plate | **14.01** | 18.12 | 15.70 | 16.20 |
| | IDP | **33.33** | 41.66 | 82.35 | 629.33 |
| Training cost [hrs], Final epoch | Cylinder | **21.41, 19** | 35.84, 21 | 64.30, 30 | 76.15, 39 |
| | Airfoil | **122.33, 39** | 176.82, 42 | 275.40, 45 | 206.55, 37 |
| | Plate | **56.07, 27** | 125.78, 19 | 176.94, 27 | 127.50, 30 |
| | IDP | **2.68E+01, 21** | 5.66E+01, 19 | 6.20E+01, 19 | NA |
| RMSE-1 [1e-2] | Cylinder | **2.04E-01** | 2.20E-01 | 2.26E-01 | 8.09E-01 |
| | Airfoil | 2.88E+01 | **2.68E+01** | 4.35E+01 | 2.93E+01 |
| | Plate | 2.87E-02 | 2.20E-02 | **1.98E-02** | 2.03E-02 |
| | IDP | **1.77E-02** | 1.87E-02 | 1.95E-02 | NA |
| RMSE-50 [1e-2] | Cylinder | **2.42** | 2.74 | 4.39 | 1.87E+01 |
| | Airfoil | **1.10E+03** | 1.22E+03 | 1.66E+03 | 1.17E+03 |
| | Plate | 3.18E-02 | **2.78E-02** | 2.88E-02 | 5.19E-02 |
| | IDP | **1.08E-01** | 3.24E-01 | 1.78E-01 | NA |
| RMSE-all [1e-2] | Cylinder | **8.37** | 8.49 | 1.07E+01 | 1.65E+02 |
| | Airfoil | **4.21E+04** | 5.56E+04 | 6.95E+04 | 6.11E+04 |
| | Plate | 1.60E-01 | **1.48E-01** | 1.51E-01 | 5.46E-01 |
| | IDP | **2.20E-01** | 3.78E-01 | 3.65E-01 | NA |

Table 1: **Detailed measurements** of our method, MS-GNN-GRID, GRAPHMESHNETS, and GRA-PHUNET. All measurements are conducted using a single Nvidia RTX 3090. BSMS-GNN consistently generates stable and competive global rollouts with the smallest training cost. BSMS-GNN is also lightweight and has the fastest inference time. It is also free from the large RMSE due to poor pooling on unseen geometries where the learnable pooling module of GRAPHUNET suffers.

**Implementation**  We implement our BSMS-GNN framework with PyTorch (Paszke et al., 2019) and PyG (PyTorch Geometric) (Fey & Lenssen, 2019). We train the entire model by supervising the single-step $L_2$ loss between the ground truth and the nodal field output of the decoding module. For more detailed information, such as the statistics of the mesh, the number of layers, the multi-edge sets, and the hyperparameters of the MLP network, please refer to Sec. A.1 and A.2. Our datasets and code are publicly available at *https://anonymous.4open.science/r/BSMS-GNN-ICLR-2023/*.

**MISCs**  We also conduct the ablation study for the specific choice of our transition method in Sec. A.3, and include the scaling test on INFLATINGFONT in Sec. A.4. Another ablation study can be performed on whether or not to use a learnable pooling module. But we already covered this aspect by comparing to GRAPHUNET in the full experiments (details in Sec. 4.2).

## 4.2  RESULTS AND DISCUSSIONS

We evaluate BSMS-GNN on all described benchmarks and compare it with the baselines (Sec. 4.1). In general, our method builds multi-level graphs without the loss of connectivity; it is free from spatial proximity and therefore avoids wrong edges across the boundary for complex geometries (the generated multi-level graphs of each example are plotted in Fig. 5), leading to high-quality rollouts on all tasks. Compared to all baselines, our method shows dominant advantages in significantly less memory footprint and training time to reach the desired accuracy, as plotted in Table. A.1.

**Disadvantages of Learnable Pooling**  Compared to other methods, GRAPHUNET has similar error only in AIRFOIL where the mesh is consistent across instances, but has significantly larger error ($2 \sim 4\times$ 1-step RMSE and $5 \sim 20\times$ rollout RMSE) in CYLINDERFLOW and DEFORMINGPLATE where instances have varying meshes. Empirically, this difference indicates that learnable modules infer poor pooling results for unseen geometries and harms information passing at coarser levels. Another concern is the overhead by adjacency enhancement. Though the learnable pooling module (Linear + Top-K) itself does not take long, GRAPHUNET needs to enhance adjacency by matrix multiplication in the forward pass. These multiplications, although reimplemented with sparse operations, result in $2 \sim 40\times$ unit training time and $4 \sim 20\times$ unit infer time (except for DEFORMING-PLATE). In INFLATINGFONT, a single epoch takes an unaffordable 50 hours, making it impossible for the full experiment. Due to these issues, we conclude that GRAPHUNET is not suitable for large-scale cases or cases with varying meshes, hence we exclude it in the further discussions.

| Case | Method | Train with different Batch # | | | | | | Infer |
|------|--------|------|------|------|------|------|------|-------|
| | | 2 | 4 | 8 | 16 | 32 | 64 | |
| Cylinder | **Our's** | **2.41** | **2.92** | **4.37** | **6.06** | **11.4** | **22.27** | **1.92** |
| | **Lino et al. (2021)** | 2.79 | 3.60 | 5.31 | 8.56 | 15.10 | - | 1.97 |
| | **Pfaff et al. (2020)** | 3.25 | 4.46 | 6.91 | 11.84 | 21.60 | - | 1.94 |
| | **Gao & Ji (2019)** | 23.33 | - | - | - | - | - | 2.18 |
| Aifroil | **Our's** | **3.66** | **5.46** | **8.88** | **15.70** | - | - | **2.02** |
| | **Lino et al. (2021)** | 4.18 | 6.25 | 10.65 | 19.25 | - | - | **2.02** |
| | **Pfaff et al. (2020)** | 5.53 | 8.90 | 16.08 | - | - | - | 2.06 |
| | **Gao & Ji (2019)** | - | - | - | - | - | - | 2.67 |
| Plate | **Our's** | **2.36** | **2.87** | **3.85** | **5.78** | **9.28** | **16.85** | 1.95 |
| | **Lino et al. (2021)** | 3.41 | 4.81 | 7.75 | 13.20 | - | - | 2.00 |
| | **Pfaff et al. (2020)** | 3.10 | 4.29 | 6.59 | 11.49 | 20.80 | - | **1.93** |
| | **Gao & Ji (2019)** | - | - | - | - | - | - | 2.18 |
| Font | **Our's** | **6.28** | **10.80** | - | - | - | - | **2.23** |
| | **Lino et al. (2021)** | 10.87 | 19.79 | - | - | - | - | 2.45 |
| | **Pfaff et al. (2020)** | 12.48 | 23.39 | - | - | - | - | 2.28 |
| | **Gao & Ji (2019)** | - | - | - | - | - | - | 4.51 |

Table 2: **Memory footprint under multi-batches**, BSMS-GNN consistently cuts RAM consumption by approximately half in all cases in the training stage, and also has the smallest (except for DEFORMINGPLATE) inference RAM.

**Unit Training/Inference Time** We evaluate the time complexity with unit training time per step. Compared to GRAPHMESHNETS, BSMS-GNN only takes 51% and 31% ∼ 39% of the unit training time for Eulerian systems (CYLINDERFLOW and AIRFOIL) and Lagrangian systems with contacts (DEFORMINGPLATE and INFLATINGFONT), respectively. The main source of the speedup is the reduction of total MP times. In GRAPHMESHNETS, 15 MP passes are conducted on the finest level of the mesh. While in our method, $2 \times$ levels $+ 3$ MPs are conducted, and only $4$ of them happen on the finest level. As for MS-GNN-GRID, they share the similar advantage of performing more MPs on smaller subsets at coarser levels, but $4 \times$ levels $+ 6$ in total MPs are required, while $8$ of which happen at the finest level; they also have the overhead of learnable aggregation/returning modules. When applied to Eulerian systems, their unit training/infer time lies between our method and GRAPHMESHNETS. For Lagrangian systems with contacts, the contact edge sets bring in additional overhead and degrade the unit training time to the same level as GRAPHMESHNETS.

Regarding inference time, the performances for DEFORMINGPLATEwith the smallest mesh size ($\sim 1$K) are very similar. Our method and MS-GNN-GRIDhave similar performance in CYLINDER-FLOW(mesh size $\sim 1.5$K) as well, and both outperformed GRAPHMESHNETS. As mesh size grows ($5$K $\sim 15$K), BSMS-GNN boost the inference time gradually up to $2.5\times$ compared to MS-GNN-GRID, and $2.9\times$ compared to GRAPHMESHNETS.

**Training time to reach desired rollout accuracy** Since rolling out is the ultimate purpose for predicting physical systems, we define the training cost (in time) as the earliest wall time to obtain the converged global rollout RMSE. The global rollout error is reduced by feeding the model with noisy inputs but correct outputs at each epoch, so that it can learn to better correct noises generated during inference (Pfaff et al., 2020). The essential point is epoch number, i.e. the number of random noise patterns seen. In our observation, all methods reach the desired global rollout RMSE with a similar amount of epochs, which leads to our superiority due to much faster unit training time.

**Accuracy** We plot the detailed RMSEs with different rollout steps (1, 50, or until the end) for different methods. Our method has the smallest global rollout RMSE for all cases except DEFORM-INGPLATE, where the error is slightly higher than the alternatives. For INFLATINGFONT, with the most complicated contact connectivities, our method cuts about $55\%$ of the training time while also reducing $40\%$ of the global rollout error.

**Memory Footprint** The memory footprint affects both the training and the inference stage. In the training stage, the higher RAM consumption sets the lower cap of batch number and results in more data transfer from CPU to GPU and a larger overhead to finish an epoch. In our observation, we can achieve at most $\sim 3$x acceleration by simply increasing the batch number. In the inference stage, the RAM consumption is closely related to the deployment in production. We measure the memory footprint of all methods under varying batch sizes (Table. 3). Compared to GRAPHMESHNETS, BSMS-GNN consistently reduces memory consumption by approximately half in all cases. As for

Figure 4: **Failure cases for MS-GNN-GRID.** Left: the configuration of the simplest failure case for multi-level GNNs by spatial proximity: the steady-state 1-D heat transfer. Right, leading two columns: two tests showing that even if trained to convergence, the erroneous edge across the boundary can still result in wrong inference. Right, last two columns: the erroneous edge coincidentally does not affect the results due to the symmetry of the solution and that no heat will diffuse between two nodes with the same temperature.

MS-GNN-GRID, we observe a similar phenomenon as the unit training time: their advantage only stands for the Eulerian systems. For the Lagrangian systems with additional contact edge sets, they consume similar or even higher memory than GRAPHMESHNETS. Overall, our method consumes $17\% \sim 57\%$ less memory than MS-GNN-GRID. Our method also has the smallest inference RAM, except for DEFORMINGPLATE where ours is slightly higher (20MBs) than GRAPHMESHNETS.

**The Failure Case for Spatial Proximity** To illustrate the adversarial impact of wrong edges built by spatial proximity, we design a simple 1-D steady-state heat transfer on sticks (Figure. 4 left). The training set contains two mirrored instances, where one end of the stick is fixed at a specific temperature, and the other has the fixed heat flux. The result of such a configuration is the linear temperature distribution. In the test set, we simply align two sticks in a head-to-tail fashion but leave some space between them so no heat diffuses across the boundary. We choose MS-GNN-GRID as an example of those utilizing spatial proximity. The training for BSMS-GNN and MS-GNN-GRID can converge quickly under a few hundred iterations. However, in the test phase, the erroneous connection by proximity transfers the information between two isolated sticks and can yield wrong results (Figure. 4 right, leading two columns). We also note that although preprocessing (separately inferring for two sticks) can help resolve the issue in this simple example, it is not doable for a single but wild geometry. The simplest counter example is the fluid dynamics in a U-shaped channel where the two ends of the channel are close spatially but far away geodesically.

## 5 CONCLUSION, LIMITATIONS, AND FUTURE WORK

Bi-Stride Multi-Scale GNN features a simple and robust pooling strategy that systematically generates an arbitrary-depth, multi-level graph neural network given geometry in the wild as the sole input. It does not rely on mesh generators or projecting to regular grids. Bi-Stride guarantees direct connections between the pooled and unpooled nodes, while free from any redundant connections by spatial proximity. This further helps replace the MLPs for the transition between adjacent levels with a unified non-parametrized transition scheme. BSMS-GNN eliminates the necessity of multiple MPs and the latent edge embedding. Combined, it significantly reducing computational costs. With moderate tailoring, BSMS-GNN can be easily extended to multi-edge-set problems involving different dynamical behaviors.

In summary, we believe that the non-parameterized Bi-Stride strategy will conceptually complete the methodology path created by GraphMeshNets, just like what striding and up-sampling by interpolation are for CNN. Following our non-parametrized strategy, there are interesting ideas to explore. For example, although any general multi-level GNN can reduce the time complexity to linear, it still need to load the whole graph initially. Combining the multi-level GNN and batch training is crucial for huge-scale graphs. Second, as stated in Li et al. (2020b), the transition from fine to coarse levels is equivalent to the transition from sparse, high-rank kernels to dense, low-rank kernels. Although the dense or fully connected graphs only appear near the bottom layers with minimum nodes in practice, there is no theoretical guarantees. Whether strategies like edge pruning is needed to avoid dense graphs at coarser levels becomes an interesting question. In addition, since all the nodal features will be smoothed without the skip-layers, how to migrate our strategy to GAE+Transformer (Han et al., 2022) is also a meaningful direction.

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

# A  APPENDIX

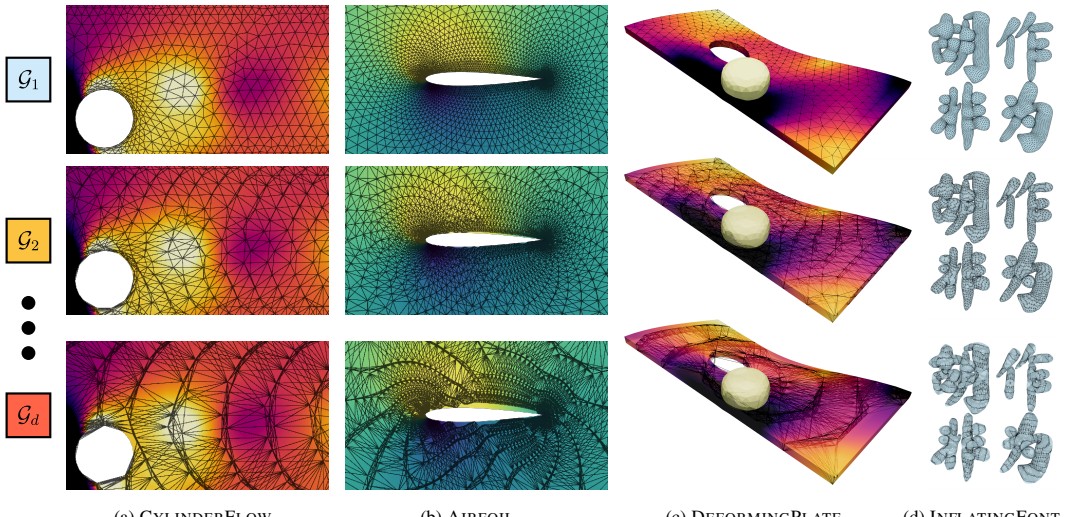

| (a) CYLINDERFLOW | (b) AIRFOIL | (c) DEFORMINGPLATE | (d) INFLATINGFONT |
|---|---|---|---|

Figure 5: **Example plots of the multi-level graphs produced by our bi-stride pooling.** Our dataset contains both Eulerian and Lagrangian systems. Many meshes are highly irregular and contain massive self-contact, which poses strong challenges for building the coarser level connection by spatial proximity. The bi-stride strategy only relies on topological information and has proven to be robust and reliable on arbitrary kinds of geometry.

## A.1  DATASET DETAILS

We adopt three existing test cases: Cylinder (Flow), Airfoil, and (Deforming) Plate from GRAPHMESHNETS. The Cylinder includes the transient incompressible flow field around a fixed cylinder at varying locations. The Airfoil includes the transient compressible flow field at varying Mach numbers around the airfoil with varying angles of attack (AOA). The Plate includes hyperelastic plates squeezed by moving obstacles. In addition to these three cases, our Font(INFLATINGFONT) case involves the quasi-static inflation of enclosed elastic surfaces (3D surface mesh) possibly with self-contact. We create the INFLATINGFONTcases using the open-source simulator (Fang et al., 2021), with the same material properties and inflation speed. The input geometries for INFLATING-FONTare $1,400\ 2\times2$-character matrices in Chinese. All the datasets are split into 1000 training, 200 validation, and 200 testing instances. In the following table, the second entries with superscript* in the average edge number column are for the contact edges:

| Case | Ave # nodes | Ave # edges | Mesh type | Seed method | # Levels | # Steps |
|---|---|---|---|---|---|---|
| Cylinder | 1886 | 5424 | triangle, 2D | MinAve | 7 | 600 |
| Airfoil | 5233 | 15449 | triangle, 2D | MinAve | 9 | 600 |
| Plate | 1271 | $4611, 94^*$ | tetrahedron, 3D | MinAve | 6 | 400 |
| Font | 13177 | $39481, 6716^*$ | triangle, 3D | CloseCenter | 6 | 100 |

Below we list the model configurations: 1) the offset inputs to prepend before the material edge processor $e_{ij}^{M}$, and $e_{ij}^{W}$, and 2) nodes $p_i$, as well as the nodal outputs $q_i$ from the decoder for each experiment cases, where $X$ and $x$ stand for the material-space and world-space positions, $v$ is the velocity, $\rho$ is the density, $P$ is the absolute pressure, and the dot $\dot{a} = a_{t+1} - a_t$ stands for temporal change for a variable $a$. All the variables involved are normalized to zero-mean and unit variance via pre-processing.

As for time integration, Cylinder, Airfoil, and Plate inherited the first-order integration from GRAPHMESHNETS. For INFLATINGFONT, the first-order quasi-static integration (Fang et al., 2021) is used in the solver. Hence, we also adopt the first-order integration for INFLATINGFONT.

| Case | Type | Offset inputs $e_{ij}^M$ | Offset inputs $e_{ij}^W$ | Inputs $p_i$ | Outputs $q_i$ |
|------|------|---------|---------|---------|---------|
| Cylinder | Eulerian | $\boldsymbol{X}_{ij}, \|\boldsymbol{X}_{ij}\|$ | NA | $\boldsymbol{v}_i, n_i$ | $\dot{\boldsymbol{v}}_i$ |
| Airfoil | Eulerian | $\boldsymbol{X}_{ij}, \|\boldsymbol{X}_{ij}\|$ | NA | $\rho_i, \boldsymbol{v}_i, n_i$ | $\dot{\boldsymbol{v}}_i, \dot{\rho}_i, P_i$ |
| Plate | Lagrangian | $\boldsymbol{X}_{ij}, \|\boldsymbol{X}_{ij}\|, \boldsymbol{x}_{ij}, \|\boldsymbol{x}_{ij}\|$ | $\boldsymbol{x}_{ij}, \|\boldsymbol{x}_{ij}\|$ | $\dot{\boldsymbol{x}}_i, n_i$ | $\dot{\boldsymbol{x}}_i$ |
| Font | Lagrangian | $\boldsymbol{X}_{ij}, \|\boldsymbol{X}_{ij}\|, \boldsymbol{x}_{ij}, \|\boldsymbol{x}_{ij}\|$ | $\boldsymbol{x}_{ij}, \|\boldsymbol{x}_{ij}\|$ | $n_i$ | $\dot{\boldsymbol{x}}_i$ |

## A.2 ADDITIONAL MODEL DETAILS

### A.2.1 BASIC MODULES AND ARCHITECTURES

The MLPs for the nodal encoder, the processor, and the nodal decoder are ReLU-activated two-hidden-layer MLPs with the hidden-layer and output size at 128, except for the nodal decoder whose output size matches the prediction $\boldsymbol{q}$. All MLPs have a residual connection. A LayerNorm normalizes all MLP outputs except for the nodal decoder.

### A.2.2 BASELINE: GRAPHMESHNETS

Our GRAPHMESHNETSimplementation uses the same MLPs as above but with an additional module: the edge encoder. Also, the edge latent is updated and carried over throughout the end of multiple MPs. We use 15 times MP for all cases to keep it consistent with GRAPHMESHNETS.

### A.2.3 BASELINE: MS-GNN-GRID

Our re-implementation of MS-GNN-GRIDuses the same MLPs as above but with four additional modules: the edge encoder at the finest level, the aggregation modules for nodes and edges at every level for the transitions, and the returning modules for nodes at every level. This method also requires assigning the regular grid nodes for each level. We assign these grid nodes by defining an initial grid resolution and an inflation rate between levels. As for the MP times at each level, we follow Lino et al. (2022a) to use four at the top and bottom levels and two for the others.

| Case | # Levels | Initial grid dx | dx inflation | Level-wise # MPs |
|------|----------|-----------------|--------------|------------------|
| Cylinder | 4 | [5e-2, 5e-2] | 2 | [4, 2, 2, 4] |
| Airfoil | 4 | [4.5, 4.5] | 2 | [4, 2, 2, 4] |
| Plate | 4 | [4e-3, 4e-3, 4e-3] | 2 | [4, 2, 2, 4] |
| Font | 4 | [1.5e-2, 1.5e-2, 1e-3] | 2 | [4, 2, 2, 4] |

### A.2.4 BASELINE: GRAPHUNET

Our re-implementation of GRAPHUNET uses the same number of levels as those of BSMS-GNN. Likewise, we make the following modifications to the original GRAPHUNET: (1) We change the information passing from GCN to our message passing module for consistency and translational invariance. (2) GraphUNet was intended for tiny graphs (100 nodes) and used dense matrix multiplications. This design is not scalable as it can break the memory limit and slow down the training to take more than 30 days per epoch in our graph size (1500 to 15000 nodes). We thus optimize the operations such as matrix multiplication and aggregation with sparse implementations.

### A.2.5 NOISE AND BATCH NUMBER

For all three methods, we enhance the datasets by shuffling noise into them so the model can resist the noise produced by single-step predictions. Each method's batch number has been tuned to achieve a good convergence rate under smaller subsets.

| Case | | Batch size | | | Noise scale |
|------|------|---------------------|---------------------|------------------|-------------|
| | Ours | Pfaff et al. (2020) | Lino et al. (2022a) | Gao & Ji (2019) | |
| Cylinder | 32 | 16 | 16 | 2 | velocity: 2e-2 |
| Airfoil | 8 | 4 | 8 | 1 | velocity: 2e-2, density: 1e1 |
| Plate | 8 | 2 | 2 | 1 | pos: 3e-3 |
| Font | 2 | 1 | 1 | 1 | pos: [5e-3, 5e-3, 3.33e-4] |

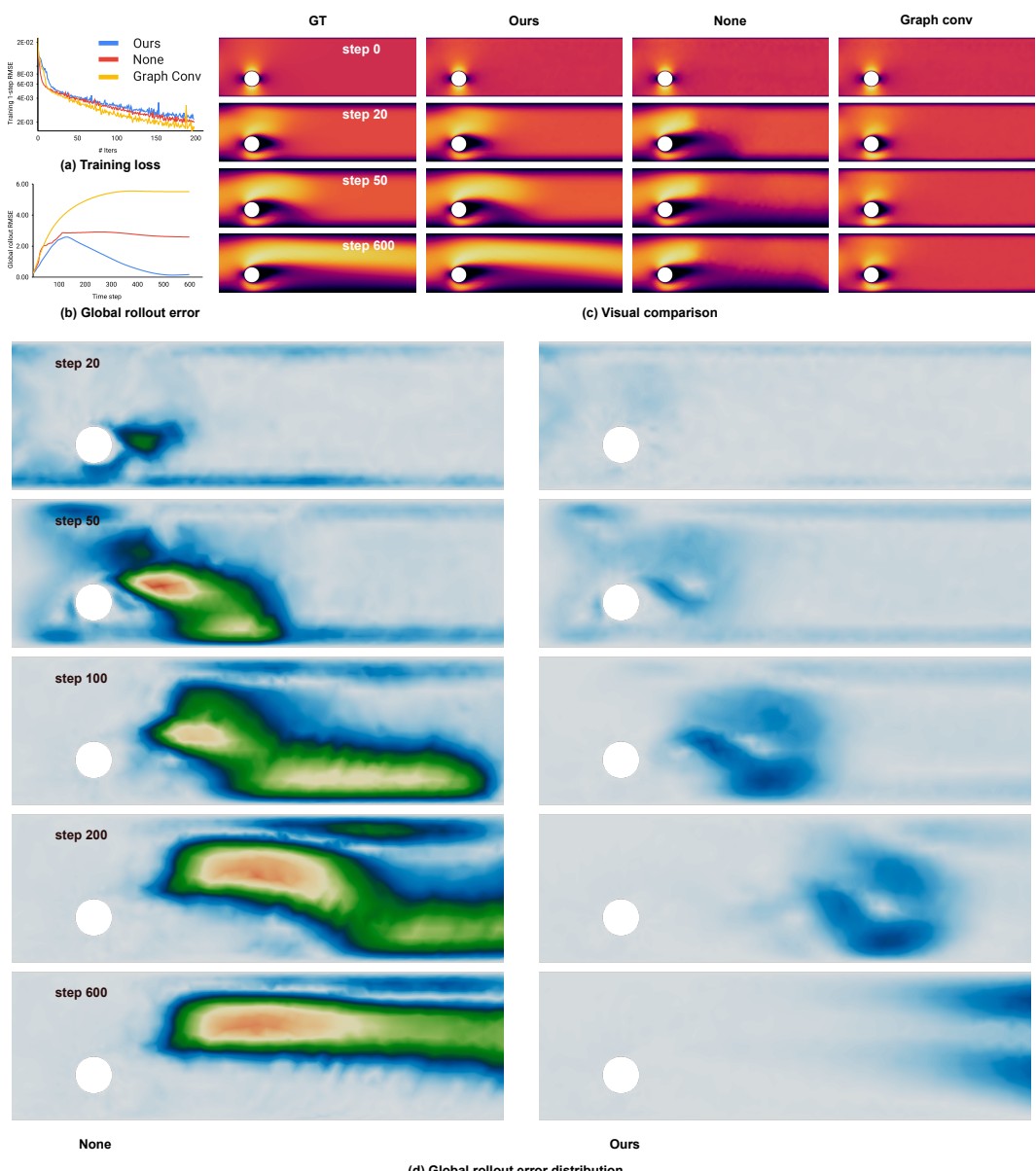

Figure 6: (a) All three transition methods can reach the target training RMSE given 200 iterations. (b) However, our weighted graph aggregration+returning has the strongest resistance to the noise during the rollout. (c) The visual comparisons show that no transition produces mosaic-like patterns, while the graph convolution transition smeared out the information and ceased propagating downstream. (d) The global rollout error distribution of no transition (**Left**) shows the edge of the mosaic patterns look similar to the simulation mesh; The error of our transition (**Right**) travels with the generated vortices downstream and leaves the domain after step 200, which explains the RMSE drop in (b).

## A.3    ABLATION STUDY

### A.3.1    TRANSITION METHOD

While exploring the non-parametric transition solutions, we started with no transition because our method is adopted directly from GUN (Gao & Ji, 2019). The no-transition strategy produces low enough 1-step RMSE and visually correct rollouts for INFLATINGFONT. However, in the global

rollouts of CYLINDERFLOWand AIRFOILcases, we observed stripe patterns (Figure. 6 (c), column **None**) where the stripes are aligned with the edges at the coarser levels (Figure. 6 (d)). We suspect that this error results from the fact that the unpooled nodes all have zero information before MP, making them indistinguishable for the processor modules and exaggerating the difference between pooled and unpooled nodes over rollouts.

The no-transition strategy resembles no interpolation during the super-resolution phase of CNN+UNet. Naturally, we then tried a single step of graph convolution (without activation) to resemble the interpolation in regular grids. However, this turns out to over-smooth the features (Figure. 6 (e), column **Graph Conv**), and the information propagation was smeared out except for the area near the generator (in this case, near the cylinder).

We believe the over-smoothing issue arises from the ignorance of the irregularity of the mesh. Unlike CNN, where the fine nodes regularly lie at the center of coarser grids, irregular meshes have varying topology and element sizes. The element sizes are almost always smaller near the interface for higher precision in simulations; hence an unweighted graph convolution can smear the finer information near the cylinder and their adjacent neighbors during returning. The natural choice to account for the irregularity is to include reasonable nodal weights (such as the size). In the end, we arrive at the solution proposed in Sec. 3.3 by utilizing the nodal weights during aggregation and recording the shares of contribution for later returning. Our transition method works consistently for all experiment cases and produces the lowest RMSE for global rollouts (Figure. 6 (b)).

**Comparing to alternative transition methods**  Additionally, we compare our transition methods to two alternatives extracted from previous works: (1) calculating the edge weights (kernel) for the information flow using the inverse of its length (node position offset), which we refer to as **Pos-Kernel** (Liu et al., 2021); and (2) the level-wise learnable transition modules implemented by additional MP, which we refer to as **Learnable** (Fortunato et al., 2022).

| Measurements | Ours | None | Graph-Conv | Pos-Kernel | Learnable |
|---|---|---|---|---|---|
| Training time/step [ms] | 10.14 | **9.30** | 10.07 | 10.06 | 17.75 |
| Infer time/step [ms] | 6.75 | **5.70** | 6.46 | 6.90 | 11.28 |
| Training RAM [GBs] | **11.041** | **11.041** | **11.041** | **11.041** | 18.033 |
| Infer RAM [GBs] | **1.923** | **1.923** | **1.923** | **1.923** | 1.931 |
| RMSE-1 [1e-2] | 2.85E-01 | **1.49E-01** | 3.41E-01 | 6.38E-01 | 4.70E-01 |
| RMSE-50 [1e-2] | 1.43E+01 | 2.05E+02 | 2.40E+02 | 1.77E+01 | **1.35E+01** |
| RMSE-all [1e-2] | 1.68E+01 | 2.59E+02 | 5.51E+02 | 2.01E+01 | **1.57E+01** |

Table 3: **Detailed measurements** of different transition methods. Ours and **Pos-Kernel** are the only two non-parametric transitions which are light-weighted and procude reliable rollouts compred to the expensive **Learnable** transition.

In addition to the high RMSE of **None** and **Graph-Conv** shown in Figure. 6, we can also observe that: (1) the training/infer time and RAM consumption for all non-parametric transitions (including **None**) are similar, which supports the statement that our transition method is light-weighted. (2) **Learnable** transition can reach slightly higher accuracy but at the price of $\sim 70\%$ more training/infer time and RAM. As mentioned in Sec. 4.2, higher training RAM can limit the batch number and increase the frequency of data communication between CPU and GPU, slowing down the training process even further when the scale goes up. (3) **Pos-Kernel** results in a slightly higher RMSE compared to our method, making it a competitive alternative choice in production.

## A.4   SCALING ANALYSIS

We train and evaluate three different methods on INFLATINGFONT with varying resolutions (5K,15K,30K, and 45K) for the scaling analysis.

**Adjustments for datasets and models**  We generate the downscale and the upscale version of INFLATINGFONT with different average node numbers for the initial geometry, and then use the same settings to simulate the sequence. As reported in Fortunato et al. (2022), the low-resolution model suffers from converging to very small RMSE; hence we loosen the termination criteria by enlarging

the target RMSE relative to the average edge length to prevent convergence failures. Similarly, the noise injection is also adjusted to be relative to the average edge length. Moreover, with a smaller number of nodes, the number of levels required to achieve the same bottom resolution also reduces. We make the corresponding adjustments to the levels of our model $d_1$ and that of the MS-GNN-GRID $d_2$. The adjustments are plotted below.

| # Nodes | $d_1$ | $d_2$ | Initial grid dx | Target RMSE | Noise in pos |
|---|---|---|---|---|---|
| 5k | 4 | 2 | [6e-2 6e-2 4e-3] | 1.73e-4 | [8.5e-3, 8.5e-3, 5.7e-4] |
| 15k | 6 | 4 | [1.5e-2 1.5e-2 1e-3] | 1e-4 | [5e-3, 5e-3, 3.33e-4] |
| 30k | 7 | 5 | [7.5e-3 7.5e-3 5e-4] | 1e-4 | [3.5e-3, 3.5e-3, 2.4e-4] |
| 45k | 7 | 5 | [7.5e-3 7.5e-3 5e-4] | 1e-4 | [2.9e-3, 2.9e-3, 1.9e-4] |

**Results**    The results in Figure. 7 show that both BSMS-GNN and MS-GNN-GRID scale up well, preserving a near-linear scale-up rate, in contrast to GRAPHMESHNETS. Still, our method is lighter weighted and more efficient than MS-GNN-GRID because of the non-parametric transitions and fewer level-wise MP.

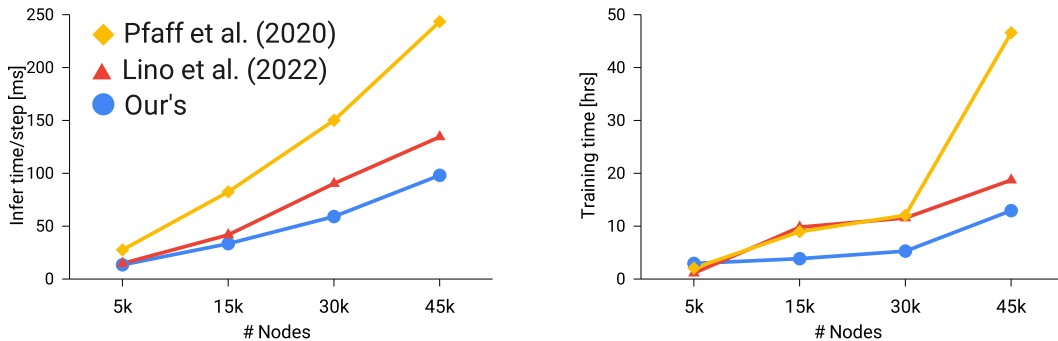

Figure 7: **Scale analysis**. With the growing size of INFLATINGFONT, BSMS-GNN shows an obvious trend of growing advantage over GRAPHMESHNETS.

## A.5    THE PROOF OF CONSERVATION OF CONTACT EDGES

With Bi-stride pooling, our pooling conserves all the contact edges under the enhancement in Eq. 2. We assume the graph is undirected and unweighted, such that the adjacent matrix is a boolean matrix.

Formally speaking, given any contact edge $(i, j)$ at level $l$ (i.e. $\boldsymbol{A}_l^C[i, j] = 1$) and a Bi-stride pooling $P$ which pools nodes $\mathcal{I}$, there exists a contact edge $(i', j')$ that remains in the coarser level (i.e. $\boldsymbol{A'}_{l+1}^C[i', j'] = 1, i', j' \in \mathcal{I}$) and $i/i', j/j'$ are connected (i.e. $\boldsymbol{A}_l[i, i'] = \boldsymbol{A}_l[j, j'] = 1$). There are only four scenarios concerning the pooling nodes $\mathcal{I}$ and the contact edge nodes $i, j$, under which the assertion always holds:

1. Both $i, j$ are pooled, i.e. $i, j \in \mathcal{I}$. Obviously $\boldsymbol{A'}_{l+1}^C[i', j'] = 1$ by letting $i' = i, j' = j$.
2. Only $i$ is pooled, $i \in \mathcal{I}, j \notin \mathcal{I}$. Since we use Bi-stride pooling, $j$ can either be the seed at level 0 (Bi-stride can select either even or odd levels) that directly connects to all nodes at level 1, or must have at least one direct connection from the previous level. I.e, at least one neighbor of $j$ in the adjacent level is pooled, we let it be $j'$: $\boldsymbol{A}_l[j, j'] = 1, j' \in \mathcal{I}$. Then $\boldsymbol{A}_l^C \boldsymbol{A}_l[i, j'] \geq \boldsymbol{A}_l^C[i, j] * \boldsymbol{A}_l[j, j'] = 1$, and $\boldsymbol{A}_l(\boldsymbol{A}_l^C \boldsymbol{A}_l)[i, j'] \geq \boldsymbol{A}_l[i, i] * (\boldsymbol{A}_l^C \boldsymbol{A}_l)[i, j'] = 1$. Let $i' = i$, then $\boldsymbol{A'}_{l+1}^C[i', j'] = 1$.
3. Only $j$ is pooled, $i \notin \mathcal{I}, j \in \mathcal{I}$. Similarly we have at least one $i'$ such that: $\boldsymbol{A}_l[i', i] = 1, i' \in \mathcal{I}$. Then $\boldsymbol{A}_l \boldsymbol{A}_l^C[i', j] \geq \boldsymbol{A}_l[i', i] * \boldsymbol{A}_l^C[i, j] = 1$, and $(\boldsymbol{A}_l \boldsymbol{A}_l^C) \boldsymbol{A}_l[i', j] \geq (\boldsymbol{A}_l \boldsymbol{A}_l^C)[i', j] * \boldsymbol{A}_l[j, j] = 1$. Let $j' = j$, then $\boldsymbol{A'}_{l+1}^C[i', j'] = 1$.
4. None of $i, j$ is pooled, $i, j \notin \mathcal{I}$. Then, we select one direct pooled neighbor for $i, j$, respectively, such that $\boldsymbol{A}_l[i', i] = \boldsymbol{A}_l[j, j'] = 1, i', j' \in \mathcal{I}$. Then $\boldsymbol{A}_l \boldsymbol{A}_l^C[i', j] \geq \boldsymbol{A}_l[i', i] * \boldsymbol{A}_l^C[i, j] = 1$, and $(\boldsymbol{A}_l \boldsymbol{A}_l^C) \boldsymbol{A}_l[i', j'] \geq (\boldsymbol{A}_l \boldsymbol{A}_l^C)[i', j] * \boldsymbol{A}_l[j, j'] = 1$.

## A.6 ALGORITHMS FOR THE SEEDING HEURISTICS

Here we elaborate our two seeding heuristics for the bi-stride pooling at every levels: picking the seed that 1) is closest to the center of a cluster (CloseCenter), and 2) with the minimum average geodesic distance to its neighbors (MinAve). The complexity for MinAve is $O(N^2)$ as we need to conduct BFS for every nodes to find the one with the minimum average distance to neighbors. In our experiments, the quadratic cost of MinAve is tolerable for all cases but INFLATINGFONT.

---

**Algorithm 1:** MinAve: seeding by minimum average geodesic distance to neighbors

---

**Input:** Unweighted, Bi-directional graph, $G = (N, E)$
**Output:** List of seeds in each clusters $L_s$
1   $L_c \leftarrow \text{DetermineCluster}(G)$
2   $L_s \leftarrow \emptyset$
    `/* BFS(s) returns the list of distances to all other neighbors from s`
        `*/`
    `/* if unreachable, the distance is set to infinity`               `*/`
3   $D \leftarrow \{\text{BFS}(s) \text{ for } s \text{ in } N\}$
4   **for** *idx in $L_c$* **do**
5       $D_c \leftarrow D[\text{idx}, \text{idx}]$
6       $\bar{D}_c \leftarrow \text{average}(D_c, \dim = 1)$
7       $s \leftarrow \text{idx}[\text{argmin}(\bar{D}_c)]$
8       $L_s.\text{append}(s)$
9   **return** $L_s$

---

For INFLATINGFONT, the largest mesh has around 47K nodes, and the time for pre-processing with MinAve becomes intolerable. We switch to CloseCenter with the linear complexity.

---

**Algorithm 2:** CloseCenter: seeding by minimum distance to the center of cluster

---

**Input:** Unweighted, Bi-directional graph, $G = (N, E)$; Positions of the nodes, $X$
**Output:** List of seeds in each clusters $L_s$
1   $L_c \leftarrow \text{DetermineCluster}(G)$
2   $L_s \leftarrow \emptyset$
3   **for** *idx in $L_c$* **do**
4       $\bar{X} \leftarrow \text{average}(X[\text{idx}], \dim = 0)$
5       $\Delta X \leftarrow X - \bar{X}$
6       $D \leftarrow ||\Delta X||_2$
7       $s \leftarrow \text{idx}[\text{argmin}(D)]$
8       $L_s.\text{append}(s)$
9   **return** $L_s$

---

For both heuristics, we conduct the search in a per-cluster fashion to avoid the information from other clusters that could pollute the search result. For example, when determining the center of an isolated part of the input geometry, the positions of nodes from other clusters could pollute this process. The determination of clusters given a graph is elaborated below.

---

**Algorithm 3:** DetermineCluster

---

**Input:** Unweighted, Bi-directional graph, $\boldsymbol{G} = (N, E)$
**Output:** List of clusters $L_c$
    /* $R$ stands for remaining nodes that are not inside any cluster    */
1   $R \leftarrow N$
2   $L_c \leftarrow \emptyset$
3   **while** $R \neq \emptyset$ **do**
4      $s \leftarrow R.\text{pop}(\ )$
5      **if** $|R| = 0$ **then**
6          $L_C.\text{append}(\{c\})$
7      **else**
8          $D \leftarrow \text{BFS}(s)$
9          $C \leftarrow \emptyset$
10        $R^* \leftarrow \emptyset$
11        **for** $n$ *in* $R$ **do**
12           **if** $D[n] = \infty$ **then**
13             $R^*.\text{append}(n)$
14           **else**
15             $C.\text{append}(n)$
16        $L_C.\text{append}(C)$
17        $R \leftarrow R^*$
18   **return** $L_c$

---

