# OpenReview forum: "Bi-Stride Multi-Scale Graph Neural Network for Mesh-Based Physical Simulation"
_ICLR.cc/2023/Conference — Submitted to ICLR 2023_

### Official Review · Reviewer_4tsk · 2022-10-23

**Confidence:** 4
**Correctness:** 3
**Technical Novelty And Significance:** 3
**Empirical Novelty And Significance:** 3
**Recommendation:** 6

**Clarity, Quality, Novelty And Reproducibility:**

This paper is easy to follow. The idea is novel in this area. The authors provide the dataset and the evaluation method involved in the experiment. So, I think it can be reimplemented.

**Strength And Weaknesses:**

The authors propose that the current limitation of GNN method for physics-based learning is pooling and building connections at coarser level. This is novel since not too many people is focused on this area. The idea is simple and easy to follow. The result shows a noticeable decreasing in training and inference time since the overhead is reduced.

There are still a few things that the author can do to improve data. First, it is better to involve some ablation study to show how each part contribute to the increase of training speed. Also, I am wondering if there are erroneous connections existed in graph produced by your method and what causes these wrong connections.

**Summary Of The Paper:**

The authors provide a new pooling strategy for GNN on learning physical systems on unstructured meshes along with a down sampling and up sampling method to reduce overhead between levels. Current GNN method to infer physical states suffers from two limitations. The computation complexity is high and over-smoothing problems will happen.  The author claim that such limitations is caused by pooling and building connection at coarser levels. So, the authors provide a new pooling strategy to solve this problem.

The motivation is to create a pooling strategy that creates a 2-nd enhancement to preserve the connectivity on any input graph. So, they topologically sort the input graph and find that pooling on every depth will create a bipartition. To resemble the bipartition in mesh, they implement the BFS to compute the distance from a seed to other nodes. Then, they will do the pooling on the BFS frontiers. Also, building auxiliary edges can benefit from the bi-stride pooling.

Also, the propose a method to reduce the overhead of transition modules between adjacent levels. There is a down sampling process to project the latent information to pivot nodes. After un-pooling, there will be an up-sampling process to return message back.

## update after reading rebuttal.
The authors resolved my concerns and I would like to keep my score.

**Summary Of The Review:**

This paper proposes a new pooling strategy for GNN in physics-based learning. They try to tackle the problem from a different aspect compared to previous research work. They borrow the idea of bi-partition, which is novel in this area. And the training speed and memory usage is decreasing, which means the computation is decreased.

---

> ### Author Response · Authors · 2022-11-10
> **1st response to reviewer 4tsK**
>
> Dear reviewer 4tsK,
>
> We sincerely thank you for carefully examining our paper and raising the requirement for more ablation studies. Combined with reviewer 2’s suggestions. We decided to fullly re-implement a prior method and add 2 ablation studies on the transitional method.
>
> Detailed response:
> 1. “Suggested experiments to add by Reviewer 2”:
>     - Gao & Ji (2019) ‘s method will be fully re-implemented because it’s strongly related. It can also experimentally show, to what extent, the loss of connectivity will impact the accuracy (related to points (6) (8) of reviewer 1).
>     - Lino et al. (2022a; 2021; 2022b): Already included.
>     - Liu et al. (2021) ‘s method can not be fully re-implemented for 2 reasons: (1) it requires pre-drawing the multi-level meshes of the input geometry, which means **manually drawing D(depth_of_multi_level) * N(size_of_database), i.e. about 5*5000=25,000 meshes**. (2) it requires prior knowledge of the PDE operator while our case is purely data-driven. Still, they propose kernel-based interpolation (non-parametric, deterministic) during the upsampling interpolation. We decide to include this relevant transition as an ablation.
>     - Fortunato et al. (2022)’s method also requires **manually drawing a massive amount of multi-level meshes** making it impractical. They used a similar learnable transition module as Lino’s work, which we decided to add as ablation for the transition method.
> 2. Concerning the erroneous connections produced by our method:
>     - This is also related to reviewer 1’s point (3) and reviewer 2’s point (3). The connection construction at the coarser levels is an ill-posed definition. But our method requires very low ordering of matrix enhancement which helps distinguish the node feature. We will emphasize this reasoning, and the experimental results compared to those using different techniques for building coarser levels in our revised paper. (Although we are unable to provide theoretical support)
>     - Reviewer 1. asked if our method will lose some connection for contact edges at coarser levels. In the revised manuscript’s appendix, we will prove that this won't occur.

---

> > ### Author Response · Authors · 2022-11-14
> > **Opensourced**
> >
> > Dear reviewers,
> >
> > We thank you again for your concern for the development of the open community.
> >
> > We now share the code via [the anonymous GitHub link](https://anonymous.4open.science/r/BSMS-GNN-ICLR-2023/). In the [ReadMe](https://anonymous.4open.science/r/BSMS-GNN-ICLR-2023/README.md), you can also find the detailed instruction on how to use the code to re-generate the RMSE we reported and the [link](https://drive.google.com/drive/folders/15UjqYdDX_Zhf-uPIs0bIZ5hYVjNiZUZy?usp=share_link) where the data and trained models are hosted.
> >
> > Thank you all, and let us know of any updates.

---

> > ### Author Response · Authors · 2022-11-15
> > **Concerning the full implementation of Gao & Ji (2019) ‘s and Tables of detailed data**
> >
> > o dear reviewers 1, 2, and 4,
> >
> > Combined your suggestions on add-on experiments and detailed data, we have made the following progress:
> >
> > 1. Again, we sincerely thank you for requiring adding a detailed table. Here we have selected the most relevant criteria, assuming our method is to be deployed in production, and plotted them in the tables below.
> > 2. We have also finished evaluating the performance of GraphUNet (Gao et al. (2019)) in all four cases (but for Font because GraphUNet is too slow).
> > 3. Although Liu et al. (2021) and Fortunato et al. (2022) are impractical for full experiments as they require drawing 20,000 meshes, we abstracted 2 **transition** methods from their work and combined them into the ablation study. We have completed this ablation study on the **transition** method as well as its influence on training speed. Please see the attached tables in this thread.
> >
> > All the related tables and descriptions will be added in the revised version before this discussion deadline.

---

> > > ### Author Response · Authors · 2022-11-15
> > > **Table for detailed results and newly finished GraphUNet (Gao et al. (2019)**
> > >
> > > |         **Measurements**         | **Case** |     **Our's**    | **Lino et al. (2022)** | **Pfaff et al. (2020)** | **Gao et al. (2019)** |
> > > |:--------------------------------:|:--------:|:----------------:|:----------------------:|:-----------------------:|:---------------------:|
> > > |      Training time/step [ms]     | Cylinder |     **10.14**    |          15.36         |          19.29          |         16.20         |
> > > |                                  |  Airfoil |     **18.82**    |          25.26         |          36.72          |         55.08         |
> > > |                                  |   Plate  |     **15.58**    |          49.65         |          49.15          |         31.88         |
> > > |                                  |    IDP   |     **45.96**    |         107.16         |          117.48         |        1,833.37       |
> > > |       Infer time/step [ms]       | Cylinder |       6.75       |        **6.18**        |          14.50          |         24.30         |
> > > |                                  |  Airfoil |     **8.64**     |          20.40         |          24.20          |         33.60         |
> > > |                                  |   Plate  |     **14.01**    |          18.12         |          15.70          |         16.20         |
> > > |                                  |    IDP   |     **33.33**    |          41.66         |          82.35          |         629.33        |
> > > | Training cost [hrs], Final epoch | Cylinder |   **21.41, 19**  |        35.84, 21       |        64.30, 30        |       76.15, 39       |
> > > |                                  |  Airfoil |  **122.33, 39**  |       176.82, 42       |        275.40, 45       |       206.55, 37      |
> > > |                                  |   Plate  |   **56.07, 27**  |       125.78, 19       |        176.94, 27       |       127.50, 30      |
> > > |                                  |    IDP   | **2.68E+01, 21** |      5.66E+01, 19      |       6.20E+01, 19      |           NA          |
> > > |           RMSE-1 [1e-2]          | Cylinder |   **2.04E-01**   |        2.20E-01        |         2.26E-01        |        8.09E-01       |
> > > |                                  |  Airfoil |     2.88E+01     |      **2.68E+01**      |         4.35E+01        |        2.93E+01       |
> > > |                                  |   Plate  |     2.87E-02     |        2.20E-02        |       **1.98E-02**      |        2.03E-02       |
> > > |                                  |    IDP   |   **1.77E-02**   |        1.87E-02        |         1.95E-02        |           NA          |
> > > |          RMSE-50 [1e-2]          | Cylinder |   **2.42E+00**   |        2.74E+00        |         4.39E+00        |        1.87E+01       |
> > > |                                  |  Airfoil |   **1.10E+03**   |        1.22E+03        |         1.66E+03        |        1.17E+03       |
> > > |                                  |   Plate  |     3.18E-02     |      **2.78E-02**      |         2.88E-02        |        5.19E-02       |
> > > |                                  |    IDP   |   **1.08E-01**   |        3.24E-01        |         1.78E-01        |           NA          |
> > > |          RMSE-all [1e-2]         | Cylinder |   **8.37E+00**   |        8.49E+00        |         1.07E+01        |        1.65E+02       |
> > > |                                  |  Airfoil |   **4.21E+04**   |        5.56E+04        |         6.95E+04        |        6.11E+04       |
> > > |                                  |   Plate  |     1.60E-01     |      **1.48E-01**      |         1.51E-01        |        5.46E-01       |
> > > |                                  |    IDP   |   **2.20E-01**   |        3.78E-01        |         3.65E-01        |           NA          |
> > >
> > > Caption: Detailed measurements between our method, MS-GNN-GRID, GRAPHMESHNETS, and GRAPHUNET. All measurements are conducted using a single Nvidia RTX 3090. BSMS-GNN consistently generates stable and competitive global rollouts with the smallest training cost. BSMS-GNNis also light-weighted and has the fastest inference time. It is also free from the large RMSE due to poor pooling on unseen geometries where the learnable pooling module of GRAPHUNET suffers. GRAPHUNET also can not scale up due to the adjacent matrix multiplication inside the forward process, making the training for Font more than 50hrs/epoch hence impossible for full experiments.

---

> > > ### Author Response · Authors · 2022-11-17
> > > **Table for ablation study on transition methods**
> > >
> > > | **Measurements**        |  **Ours**  |  **None**  | **Graph-Conv** | **Pos-Kernel** | **Learnable-GMP** |
> > > |-------------------------|:----------:|:----------:|:--------------:|:--------------:|:-----------------:|
> > > | Training time/step [ms] |    10.14   |  **9.30**  |      10.07     |      10.06     |       17.75       |
> > > | Infer time/step [ms]    |    6.75    |  **5.70**  |      6.46      |      6.90      |       11.28       |
> > > | Training RAM [GBs]      | **11.041** | **11.041** |   **11.041**   |   **11.041**   |       18.033      |
> > > | Infer RAM [GBs]         |  **1.923** |  **1.923** |    **1.923**   |    **1.923**   |       1.931       |
> > > | RMSE-1 [1e-2]           |    0.29    |  **0.15**  |      0.34      |      0.64      |        0.47       |
> > > | RMSE-50 [1e-2]          |    14.30   |   205.00   |     240.00     |      17.70     |     **13.50**     |
> > > | RMSE-all [1e-2]         |    16.80   |   259.00   |     551.00     |      20.10     |     **15.70**     |
> > >
> > > Caption: Detailed measurements of different transition methods. Our’s and Pos-Kernel(from Liu et al. (2021)) are the only two non-parametric transitions that are light-weighted and produce reliable rollouts compared to the expensive Learnable transition(from Fortunato et al. (2022)).

---

### Official Review · Reviewer_YsGt · 2022-10-24

**Confidence:** 2
**Correctness:** 3
**Technical Novelty And Significance:** 2
**Empirical Novelty And Significance:** 2
**Recommendation:** 3

**Clarity, Quality, Novelty And Reproducibility:**

With the current description, I do not feel able to re-implement the algorithm. Due to the many small bugs in the explanation, I find the paper unclear and hard to follow.

**Strength And Weaknesses:**

* The problem considered in this paper seems to be very relevant.

* The experimental evaluation seems compelling as the authors can demonstrate results superior to SOTA. The caveat here is that the algorithms are compared on simulated data only although that is common practice in that particular field.

* "works on any challenging mesh in the wild": Considering that the authors evaluate on simulated data, this is a bold claim and should be removed.

* My objections against this paper stem from the description of the algorithm. Generally, there are so many inaccuracies (each one of them minor by itself) that I have difficulties understanding the proposed solution.
In details:
2
"Motivations of Our Method": What is $A$? The definition (adjacency matrix) only comes in 3.3. This makes reading unnecessary speculative.
What are "pivot nodes"?
What does connection conservatism mean? Why is this a good metric to optimize?
Fig 2: What is the update step? What is  an MP?
3
edge set: Are edges discrete or continuous?
Eq (1): Where does $v_0,i$ come from?
3.2
What is 2nd order enhancement?
What is a BFS frontier?
What does "stride and pool all nodes at every other BFS frontiers" mean? Why is that a bi-stride?
Seeding heursitics: What is that and why is it relevant for this problem?
What is "interfacial information"?
How can "two adjacent matrices $A_l$ and $A_l^C$ be determined? What is $C$? Is it related to the edge weights in $C$ in "Downsampling" in 3.3?
3.3
The authors claim to "reduce the overhead of the learnable transition modules" but do not provide insight as to why that is the case.
What is a "conserved variable"?
What is the nodal mass?
What is a near uniform mesh?
What are the volume/mass fields for irregular meshes? How can that be determined?
Upsampling: Why do all nodes except the pivots have zero information after unpooling?

It could be that all those minor issues are clear to an expert. Nonetheless, the paper needs a thorough re-write to avoid guesswork and speculation.

**Summary Of The Paper:**

This paper is concerned with how to pool irregularly spaced meshes for multi-scale estimation in physical simulations. The idea proposed herein is to use so-called bi-strides which "pools nodes by striding every other BFS frontier". The importance lies in the fact that many meshes of important problem exhibit complex topologies so naive pooling strategies may fail.

**Summary Of The Review:**

see above

---

> ### Author Response · Authors · 2022-11-10
> **1st Response to reviewer YsGt**
>
> Dear reviewer YsGT,
>
> We thank you for your time spent examining our work and raising concerns about the difficulty for a broader group of readers to comprehend our work.
>
> We have carefully examined all of your confusion and will adopt the relevant clarification to make this work more accessible to readers.
>
> We provide detailed explanations and references here for improving the clarity of our algorithm and the related terms used in the paper. We are also happy to include these details in the final paper. We sincerely hope you will reconsider our work during the discussion phase. The code, trained models, and dataset (with multi-level mesh) will also be shared before the closed window of this discussion. We will send another comment when we upload them
>
> The general graph knowledge explanations:
>
> 1. “Edges discrete or continuous?”: The edge of a graph is discrete and represents the connectivity between two nodes. Usually, it’s written in the form of a tuple of integers e=(i,j), which means nodes i and j are connected.
> 2. Adjacent matrix: a boolean (i.e. element value is either 0 or 1) matrix of size n*n, where n is the number of nodes. For every existing edge of graph e=(i,j), A(i,j)=1; all remaining elements are 0. The adjacent matrix can hence be seen as a form of representing the graph connectivity as it marks all edges with 1.
> 3. BFS: breadth-first-search; BFS frontier: the last level of nodes expanded during BFS.
> Seeding heuristic and why it's relevant. BFS needs a starting level of nodes(seeds), which can be arbitrarily chosen and raises the need for a reasonable heuristic depending on the problem.
> 4. Stride: Stride is similar to slicing. For example, if we say stride on 1,3,5,7,.. levels, we mean that we select and pool the nodes on 1,3,5,7… levels.
> 5. Bi-stride: The bi- prefix means “two”. The bi-stride name comes from a well-known concept of bi-partition of a bi-partite graph, where a graph can be split into two partitions. Here bi-stride can also be understood as striding every two BFS levels and putting them into two groups: the odd levels (1,3,5,7…) and even levels (0,2,4,6…). Selecting either group is a bi-stride pooling.
> 6. Interfacial information: this means the information passed between the common interface of 2 separate systems. For example, two elastic bodies are bouncing into each other where the contact force is the interfacial information.
> 7. "Reduce the overhead of the learnable transition modules" but do not provide insight. We thank you for requiring rigorous reasoning. But if a module only needs less than 10 deterministic FLOPS, and does not need training, it apparently requires less computational resources compared to training or inferring a neural network counterpart.
> 8. “Conserved quantities; discrete fields on the mesh”
>     - “Conserved variable”: Here we intended to use the term “conserved quantity (not variable)” in physics, such as momentum and energy. The projection of these quantities between multi-level has been extensively studied in physics-based simulations, from which we drew some intuition. We thank you for pointing out the confusion and typo here.
>     - “What is the mass/volume field on mesh, and how can that be determined”: the mass/volume field on the mesh can be calculated by iterating through each element of the mesh: (1) calculating the element’s mass/volume, (2) and evenly splitting this element’s mass/volume to its nodes. In all, this is a deterministic pre-process.
>
>
> **To be continued**

---

> > ### Author Response · Authors · 2022-11-10
> > **Continue response 1**
> >
> > Conituned:
> >
> > Other responses are listed below:
> >
> > 1. “The confusions from abbreviations”: We will re-allocate the ordering so that all abbreviations come with definitions at their 1st appearances.
> > 2. “The confusions in the schematic pipeline”:
> >     - We will use consistent descriptions. Eg, “pivot nodes” will be replaced with “pooled nodes”.
> >     - Connection conservation is illustrated in Sec. 1. and Sec. 3.2. as “not introducing partition into the graph”. The reason for this being an (empirically) good metric is mentioned in Sec.1. (page 2, middle) as “..., which stops information exchange across the separated clusters”.
> >     - The update step’s definition is contained in Fig 2. itself, such that the physical state is pushed forward in 1-time step. MP is an abbreviation for message-passing. We will include this in the caption, or just use “MessagePassing” in a non-abbreviation style.
> > 3. “The confusion from the equation related to graph knowledge”
> >     - "v0,i definition in Eq. 1.": We didn't include v0 in Eq. 1., but the subscript $_i, _j$ refers to integers representing the index of nodes. Please see the above graph knowledge explanations (1).
> >     - Kth order enhancement has been defined in Sec.2. (page 3, middle). The 2nd order is simply by setting K=2.
> >     - Adjacent matrices determination: The adjacent matrix of the original mesh is constructed by the given mesh connectivity (see above graph knowledge explanations 1. b). The adjacent matrix of the auxiliary edge set is by (1) determining the connectivity by spatial proximity as described on (page 5, bottom), then (2) setting that to matrix form (see above explanation 1. b).
> >     - The $A^C$'s superscript $^C$ was intended for **contact** edges since in our datasets the only source of interfacial information is **contact** between bodies. We can rewrite it with a better notation.
> > 4. “Upsampling: Why do all nodes except the pivots have zero information after unpooling”. We thank you for pointing out this confusion. Indeed there is no strict definition that unpooling is accompanied or not with an interpolation. However, in many previous works, such as Gao & Ji (2019) or many CNN papers, the unpooling (copying the coarser level information to the proper entries of the fine level information tensor, while leaving other entries zero) and interpolation (the other zero entries) are defined two processes. This definition, by construction, leads to our statement above. We will illustrate our definition more clearly.

---

> > > ### Author Response · Authors · 2022-11-14
> > > **Opensourced**
> > >
> > > Dear reviewers,
> > >
> > > We thank you again for your concern for the development of the open community.
> > >
> > > We now share the code via [the anonymous GitHub link](https://anonymous.4open.science/r/BSMS-GNN-ICLR-2023/). In the [ReadMe](https://anonymous.4open.science/r/BSMS-GNN-ICLR-2023/README.md), you can also find the detailed instruction on how to use the code to re-generate the RMSE we reported and the [link](https://drive.google.com/drive/folders/15UjqYdDX_Zhf-uPIs0bIZ5hYVjNiZUZy?usp=share_link) where the data and trained models are hosted.
> > >
> > > Thank you all, and let us know of any updates.

---

### Official Review · Reviewer_iDNU · 2022-10-25

**Confidence:** 3
**Correctness:** 3
**Technical Novelty And Significance:** 4
**Empirical Novelty And Significance:** 2
**Recommendation:** 6

**Clarity, Quality, Novelty And Reproducibility:**

The writing in the paper is of high quality, with a good flow and minimal typos. Logic is easy to follow, and methods are explained clearly.

The solution to the proposed problem is elegant and appears to be original.

------------------------------------------------------

Typos / Clarity issues

"BFS" (acronym not yet defined),  abstract

"pivot nodes" (not defined), p3

"none-parameterized transition", p3

 "$\{p_i, q_i\}$" (should be "$\{p^i, q^i\}$" for consistent notation), p4

"$\forall s\in[1, S]$" (implies $s$ can be any real number in $[1,S]$), p4

**Strength And Weaknesses:**

------------------------------------------------------

Strengths

The paper is well written and addresses a relevant problem. Previous methods that attempt to solve this problem are well-documented. The solution is elegant and is shown to be quite effective at significantly improving efficiency relative to the considered competitors.

------------------------------------------------------

Weaknesses

The main weakness are in the experiments:

1. Not enough competitors are considered. Several previous methods for graph pooling are mentioned:

- Gao & Ji (2019)
- Lino et al. (2022a; 2021; 2022b)
- Liu et al. (2021)
- Fortunato et al. (2022)

However, only the work of Lino, et al. is compared in the experiments. It is important that the test error and efficiency for all methods be compared.

2. For the considered methods, there is no report of the test error from the converged models. Although BSMS-GNN is shown to achieve the same test error as GraphMeshNets as reported by Pfaff et al., Lino et al. (Simulating Continuum Mechanics with Multi-Scale Graph Neural Networks) report greater performance for MS-GNN-Grid than than GraphMeshNets. Thus, it is not clear which method produces the best test error.

3. Related to points 1 and 2, it is not shown experimentally that connections that violate the boundary necessarily result in worse performance.

4. It is claimed that BSMS-GNN will work well on the cloth benchmarks. This claim is too strong to be made without experimental evidence since the dynamics of the cloth appear to be very different from those considered.



**Summary Of The Paper:**

Long-range spatial information can be passed with a low computational cost through pooling and unpooling operations. On an irregular mesh, there is a challenge of how to select pivot nodes and assign non-pivot nodes to a pivot node for pooling so that all assignments are at most 2 hops away. Previous methods have used spatial information for assignment, but this can result in assignments of nodes greater than 2 hops away or even for which a path does not exist.

This paper makes two contributions:

1. A pooling strategy for pivot node selection and non-pivot node assignment that guarantees all non-pivots are assigned to a pivot at most 2 hops away.

2. A deterministic, parameter-free pooling/ unpooling algorithm

BSMS-GNN is compared to 2 competitors: one is a flat GNN, and the other is a hierarchical GNN that uses spatial information for pooling. The comparison is made on 4 datasets for physical simulation. BSMS-GNN is shown to significantly improve number of training epochs to reach a target test error, time per epoch, inference time, and memory requirements on all datasets. It's also shown on a small toy dataset that the hierarchical competitor adds incorrect edges, while BSMS-GNN does not.


**Summary Of The Review:**

This contribution is marginally above the acceptance threshold. It provides an elegant solution to an important problem with the bi-stride pooling strategy and proposes a parameter-free downsampling technique that allows for deeper hierarchies. This contribution could be strengthened with more thorough experiments.

---

> ### Author Response · Authors · 2022-11-10
> **1st Response to reviewer iDNU**
>
> Dear reviewer iDNU,
>
> We sincerely thank you for carefully examining our paper and raising the requirement for more quantitative experiments. We analyzed your recommendation for new comparisons and decided to fully implement 1 more prior method and add 2 more ablation studies, the reason for such selections is discussed in the detailed response below.
>
> We will also examine the typos and inconsistent expressions that you so helpfully pointed out.
>
> For bullet point (1) in your review, we want to clarify that our strategy guarantees all non-pivot nodes are at most 1-hop (not 2-hops) away from pivot nodes. Please refer to Sec. 3.2. of our original paper.
>
> Detailed response:
> 1. “Suggested experiments to add”:
>     - Gao & Ji (2019) ‘s method will be fully re-implemented because it’s strongly related. It can also experimentally show, to what extent, the loss of connectivity will impact the accuracy (related to points (6) (8) of reviewer 1).
>     - Lino et al. (2022a; 2021; 2022b): Already included.
>     - Liu et al. (2021) ‘s method can not be fully re-implemented for 2 reasons: (1) it requires pre-drawing the multi-level meshes of the input geometry, which means **manually drawing D(depth_of_multi_level) * N(size_of_database), i.e. about 5*5000=25,000 meshes**. (2) it requires prior knowledge of the PDE operator while our case is purely data-driven. Still, they propose kernel-based interpolation (non-parametric, deterministic) during the upsampling interpolation. We decide to include this relevant transition as an ablation.
>     - Fortunato et al. (2022)’s method also requires **manually drawing a massive amount of multi-level meshes** making it impractical. They used a similar learnable transition module as Lino’s work, which we decided to add as ablation for the transition method.
> 2. “Report converged model’s performance”: We thank you for mentioning the relevant figures that we forgot to report, although our method also has advantages in several aspects. We will include these details in new standalone tables (also mentioned in Reviewer 1’s point (1)) for clearer reference.
> 3. “Violating the boundary(crossing boundary connections) will hurt accuracy?”: We thank you for mentioning this concern. Indeed, creating cross-boundary edges falls in the category of topological pollution, which theoretically will harm the accuracy. But this problem is not well studied even in more general GNN cases. Hence, we cannot provide theoretical error bound. Experimentally, we will discuss the comparison between our method and Lino’s, and Gao’s, as well as the ablation study for Liu’s transition method (which uses spatial proximity) in the revised version. Finally, we will explicitly admit this limitation.
> 4. “Claim that our method can work on an untested case, cloth simulation, is too ambitious”: Thank you. We will remove this statement in the revised version.

---

> > ### Author Response · Authors · 2022-11-14
> > **Opensourced**
> >
> > Opensourced
> >
> > Dear reviewers,
> >
> > We thank you again for your concern for the development of the open community.
> >
> > We now share the code via [the anonymous GitHub link](https://anonymous.4open.science/r/BSMS-GNN-ICLR-2023/). In the [ReadMe](https://anonymous.4open.science/r/BSMS-GNN-ICLR-2023/README.md), you can also find the detailed instruction on how to use the code to re-generate the RMSE we reported and the [link](https://drive.google.com/drive/folders/15UjqYdDX_Zhf-uPIs0bIZ5hYVjNiZUZy?usp=share_link) where the data and trained models are hosted.
> >
> > Thank you all, and let us know of any updates.

---

> ### Author Response · Authors · 2022-11-10
> **Detailed tables, new experiments and ablations**
>
> To dear reviewers 1, 2, and 4,
>
> Combined your suggestions on add-on experiments and detailed data, we have made the following progress:
>
> 1. Again, we sincerely thank you for requiring adding a detailed table. Here we have selected the most relevant criteria, assuming our method is to be deployed in production, and plotted them in the tables below.
> 2. We have also finished evaluating the performance of GraphUNet (Gao et al. (2019)) in all four cases (but for Font because GraphUNet is too slow).
> 3. Although Liu et al. (2021) and Fortunato et al. (2022) are impractical for full experiments as they require drawing 20,000 meshes, we abstracted 2 **transition** methods from their work and combined them into the ablation study. We have completed this ablation study on the **transition** method as well as its influence on training speed. Please see the attached tables in this thread.
>
> All the related tables and descriptions will be added in the revised version before this discussion deadline.

---

> > ### Author Response · Authors · 2022-11-15
> > **Table for detailed results and newly finished GraphUNet (Gao et al. (2019)**
> >
> >
> > |         **Measurements**         | **Case** |     **Our's**    | **Lino et al. (2022)** | **Pfaff et al. (2020)** | **Gao et al. (2019)** |
> > |:--------------------------------:|:--------:|:----------------:|:----------------------:|:-----------------------:|:---------------------:|
> > |      Training time/step [ms]     | Cylinder |     **10.14**    |          15.36         |          19.29          |         16.20         |
> > |                                  |  Airfoil |     **18.82**    |          25.26         |          36.72          |         55.08         |
> > |                                  |   Plate  |     **15.58**    |          49.65         |          49.15          |         31.88         |
> > |                                  |    IDP   |     **45.96**    |         107.16         |          117.48         |        1,833.37       |
> > |       Infer time/step [ms]       | Cylinder |       6.75       |        **6.18**        |          14.50          |         24.30         |
> > |                                  |  Airfoil |     **8.64**     |          20.40         |          24.20          |         33.60         |
> > |                                  |   Plate  |     **14.01**    |          18.12         |          15.70          |         16.20         |
> > |                                  |    IDP   |     **33.33**    |          41.66         |          82.35          |         629.33        |
> > | Training cost [hrs], Final epoch | Cylinder |   **21.41, 19**  |        35.84, 21       |        64.30, 30        |       76.15, 39       |
> > |                                  |  Airfoil |  **122.33, 39**  |       176.82, 42       |        275.40, 45       |       206.55, 37      |
> > |                                  |   Plate  |   **56.07, 27**  |       125.78, 19       |        176.94, 27       |       127.50, 30      |
> > |                                  |    IDP   | **2.68E+01, 21** |      5.66E+01, 19      |       6.20E+01, 19      |           NA          |
> > |           RMSE-1 [1e-2]          | Cylinder |   **2.04E-01**   |        2.20E-01        |         2.26E-01        |        8.09E-01       |
> > |                                  |  Airfoil |     2.88E+01     |      **2.68E+01**      |         4.35E+01        |        2.93E+01       |
> > |                                  |   Plate  |     2.87E-02     |        2.20E-02        |       **1.98E-02**      |        2.03E-02       |
> > |                                  |    IDP   |   **1.77E-02**   |        1.87E-02        |         1.95E-02        |           NA          |
> > |          RMSE-50 [1e-2]          | Cylinder |   **2.42E+00**   |        2.74E+00        |         4.39E+00        |        1.87E+01       |
> > |                                  |  Airfoil |   **1.10E+03**   |        1.22E+03        |         1.66E+03        |        1.17E+03       |
> > |                                  |   Plate  |     3.18E-02     |      **2.78E-02**      |         2.88E-02        |        5.19E-02       |
> > |                                  |    IDP   |   **1.08E-01**   |        3.24E-01        |         1.78E-01        |           NA          |
> > |          RMSE-all [1e-2]         | Cylinder |   **8.37E+00**   |        8.49E+00        |         1.07E+01        |        1.65E+02       |
> > |                                  |  Airfoil |   **4.21E+04**   |        5.56E+04        |         6.95E+04        |        6.11E+04       |
> > |                                  |   Plate  |     1.60E-01     |      **1.48E-01**      |         1.51E-01        |        5.46E-01       |
> > |                                  |    IDP   |   **2.20E-01**   |        3.78E-01        |         3.65E-01        |           NA          |
> >
> > Caption: Detailed measurements between our method, MS-GNN-GRID, GRAPHMESHNETS, and GRAPHUNET. All measurements are conducted using a single Nvidia RTX 3090. BSMS-GNN consistently generates stable and competitive global rollouts with the smallest training cost. BSMS-GNNis also light-weighted and has the fastest inference time. It is also free from the large RMSE due to poor pooling on unseen geometries where the learnable pooling module of GRAPHUNET suffers. GRAPHUNET also can not scale up due to the adjacent matrix multiplication inside the forward process, making the training for Font more than 50hrs/epoch hence impossible for full experiments.

---

> > ### Author Response · Authors · 2022-11-17
> > **Table for ablation study on transition methods**
> >
> >
> > | **Measurements**        |  **Ours**  |  **None**  | **Graph-Conv** | **Pos-Kernel** | **Learnable-GMP** |
> > |-------------------------|:----------:|:----------:|:--------------:|:--------------:|:-----------------:|
> > | Training time/step [ms] |    10.14   |  **9.30**  |      10.07     |      10.06     |       17.75       |
> > | Infer time/step [ms]    |    6.75    |  **5.70**  |      6.46      |      6.90      |       11.28       |
> > | Training RAM [GBs]      | **11.041** | **11.041** |   **11.041**   |   **11.041**   |       18.033      |
> > | Infer RAM [GBs]         |  **1.923** |  **1.923** |    **1.923**   |    **1.923**   |       1.931       |
> > | RMSE-1 [1e-2]           |    0.29    |  **0.15**  |      0.34      |      0.64      |        0.47       |
> > | RMSE-50 [1e-2]          |    14.30   |   205.00   |     240.00     |      17.70     |     **13.50**     |
> > | RMSE-all [1e-2]         |    16.80   |   259.00   |     551.00     |      20.10     |     **15.70**     |
> >
> > Caption: Detailed measurements of different transition methods. Our’s and Pos-Kernel(from Liu et al. (2021)) are the only two non-parametric transitions that are light-weighted and produce reliable rollouts compared to the expensive Learnable transition(from Fortunato et al. (2022)).

---

### Official Review · Reviewer_fbaN · 2022-10-25

**Confidence:** 4
**Correctness:** 3
**Technical Novelty And Significance:** 3
**Empirical Novelty And Significance:** 2
**Recommendation:** 5

**Clarity, Quality, Novelty And Reproducibility:**

The paper is clear and easy to follow. The references are fair and sufficient. The method is original and the claims are well supported in the experimental part. However, l am not sure if the paper is reproductible as long as it is not mentioned if the code will be provided.

**Strength And Weaknesses:**

The strength of the paper is in the novel design of the inter-scale pooling strategy that preserves the complex geometrical structure of a mesh.  It is not an easy problem in physics tasks because the downsampling / upsampling operations need to respect the mesh topology including the faces and cells. This is also related to the design of mesh and type of triangulations (tetraether, triangle, rectangle, etc) at the different parts of the physical domain. The developed strategy is extensively compared with state-of-the-art methods and shows an important gain in the training complexity, inference time, and memory footprint.

However, a set of points are not discussed :

1/ Absence of quantitative results (in a table) and comparison with state-of-the-art methods including some ablation studies on pooling strategy. We need to assess how the model performs. This point is of high importance.

2/ A baseline comparison with deterministic pooling is missing. The idea is to see to what extent is important to have a learnable pooling rather than setting à priori the different adjacency matrices at different scales and the transition matrices between two successive scales given the physical knowledge we have on the task and that the finest-grained mesh is known. In other terms,  One can predefine the structure of pooling in a deterministic manner given the physical domain boundary and geometry. Local connectivity is easily achieved and doesn't let the downsampling make connected-components (avoid isolated nodes).

3/ Why do you think you need to keep the graph mesh structure? What about training a model on point clouds such as PointNet, PointNet++, and Geodesic convolution where the graphs at different scales are constructed using a radius graph sampler? These models have shown good results in neighboring tasks and can help to get rid of the graph/mesh. Especially, in physics, there is no unique graph for a downstream task. Graph/mesh construction is an ill-posed problem. To confirm / infirm that a comparison between a cloud of points methods VS Graph methods is needed. This point is a follow-up of 1/ and 2/.

4/ Is your model impacted by the over-smoothness, and-oversquashing problems encountered in GNNs?

5/ In section 2 "background and related works", challenges in GNNs are not discussed. It could be helpful to add a paragraph to discuss the current challenges in GNNs including over-smoothness, and over-squashing w.r.t to spatial and spectral methods while making a link with the bi-stride neural network.

6/ In figure one,  you said that it can lead to a wrong connection ? one simple way is to use PointNet and iterate with a neural network sufficiently deep to propagate the information from all the points to all the points in the coarse level. It ensures the global representation of information by local propagation in a hierarchical way which can help to keep the physical structure of data. It is linked to 3/

7/ You mentioned in the motivation that a loss of connectivity is observed even with a powered adjacency matrix (higher-order). From an over-smoothness standpoint, higher-order adjacency matrices lead to a drop in performance. It is related to the fact that to ensure that the information has reached all the points, the depth of the neural network needs to be equivalent to the diameter of the graph (which is not tractable in practice). Moreover, higher-order adjacency matrices as the depth increase lead to A=1 at every entry. As a consequence the nodes become indistinguishable.

8/ What is wrong if pivot and non-pivot nodes are not connected? Can you elaborate on that and add more information to the paper?

9/ The claim of 3.2 Bi-stride pooling and adjacency enhancement: 2) not introducing wrong edges by spatial proximity. Why not?
Extra edges beyond the local neighboring system could help to propagate information to further nodes which could not be accessible due to the over-smoothness of GNN architectures. It is related to 7/

10/ Can you support your claim? "This enhancement can be geometrically interpreted as such: an auxiliary edge (i, j) should exist if j is reachable from i in 2 hops and one of which is an auxiliary edge at the finer level. "

11/ In A.4 scaling analysis, what is the graph-conv used in figure 7 ? and why not compared with GraphMeshNet and other related methods?






**Summary Of The Paper:**

The paper tackles the problem of pooling at different scales in the context of irregular geometries with a different level of granularity. The challenge is to learn / design the transition matrix between two successive scales while preserving the topological structure of the physics of the underlying systems. This strategy could be beneficial to study efficiently physical systems which are described by graph-meshes. The latter are highly irregular, especially at the boundary layer.


**Summary Of The Review:**

The paper is good but several points need to be clarified and adrressed. I can increase my score depending on the responses and the updates.

---

> ### Author Response · Authors · 2022-11-10
> **1st Respons to Reviewer fbaN**
>
> Dear reviewer fbaN,
>
> We sincerely thank you for your careful reading and insightful suggestions for our work. Bullet points 5) 7) can directly help improve the illustration of this paper; we will incorporate them into our next vision. And points 3) 6) have so helpfully suggested interesting prior works to compare to (if relevant) that were previously not in our domain knowledge.
>
> You raised concerns about reproducibility. We're sorry that we didn't have the code ready and uploaded at the time of the submission. The code, trained models, and dataset (with multi-level mesh) will be shared within this discussion window. We will send another comment when this is done.
>
> For bullet point (2) in your review, we actually used deterministic pooling to save the computational overhead, which is one of our contributions. So we believe that comparing to deterministic pooling is unnecessary as we are not using learnable poolings.
>
> The detailed responses are listed below:
>
> 1. “Detailed numbers and tables”: We will add the detailed tables listing all comparison data in the Appendix in the revision.
> 2. “Compare with deterministic pooling”: We actually use deterministic pooling (not learnable pooling) to save computational overhead. Combined with Reviewers 2 and 4’s suggestions, we decided to conduct an ablation study to compare with previous learnable pooling/transition methods.
> 3. “Use non-structure method to build connections or to build coarser levels; such as Pointnet etc.”
>     - Pure/flat point cloud was reported to have lower performance in the GraphMeshNet paper. And they have concluded that it’s the loss of the original connectivity that leads to the performance drop. As such, we decide to exclude this comparison from the present study.
>     - Using point clouds and sample radius (ball) to create a multi-level structure is somewhat the same as in Lino’s which is included in our comparisons; Only that they do not use a sphere but a cell(cube) to sample neighboring nodes.
> 4. “Can our method overcome Oversmoothing?”: We cannot theoretically prove this statement. However, we can experimentally argue this by comparing our results with the flat GNN where more MPs are needed.
> 5. “Lack a transition on the current challenge of flat GNN”: Thank you for mentioning this which can be a perfect transition in the introduction, we will add the corresponding discussion in Section 2.
> 6. “Passing global information by MPs on a coarser level between all nodes”: Yes, your observation of a fully connected graph is precise. In our experiment, the bottom level is usually fully connected. Concerning using point cloud and sample radius, please check response (3).
> 7. “An alternative view of adjacency matrix enhancement and why it may lead to over-smoothing”: Thank you for providing such an insightful alternative view that we were previously unaware of. We will merge this point of view in our revised paper.
> 8. “Why pivot and non-pivot nodes have to be connected”:
>     - Thanks for pointing out the lack of reasoning here. The lack of direct connection between the pivot and non-pivot nodes may cause the loss of connectivity for adjacent pivot nodes in the next level, hence reducing the information passing between them. However, this falls in the category of topological pollution, where we can hardly find a rigorous bound for the error caused by the pollution. But we will empirically argue this in the revised version and also provide experiments (see response 8. (b)).
>     - For experiment support, we decide to add the re-implementation of Gao & Ji (2019) ‘s method since it uses a learnable pooling module and has no guarantee of connectivity conservation. The updated results will come with the revised paper during this discussion window.
> 9. “Build coarser between nodes purely based on proximity?”: There are scenarios where being spatially close doesn't mean being geodesically close. In fact, nodes that are topologically disconnected may be spatially nearby. Our simple heat transfer experiment in Sec. 4.2 (Fig. 6.) shows this scenario. For general cases, using spatial proximity may cause topology pollution which is discussed in our response (8).
> 10. “Can you prove the correctness of building contact edges at coarser levels”: Thanks. We will prove this in the revised appendix.
> 11. “What does graph-conv mean in Fig. 7. Why not do ablation by combining this with previous techniques”
>     - The graph-conv means a standard unweighted graph convolution without activation. We will make this clear in the revised version.
>     - Fig. 7. is showing that under our framework, some transition methods will cause global rollout errors, justifying our choice of the transition. In prior techniques, they either do not have the transition or have their own learnable/kernelized transitions. Combined with Reviewers 2 and 4’s suggestions, we decided to conduct an ablation study to compare with previous learnable pooling/transition methods (also mentioned in response (2)).

---

> > ### Author Response · Authors · 2022-11-14
> > **Opensourced**
> >
> > Dear reviewers,
> >
> > We thank you again for your concern for the development of the open community.
> >
> > We now share the code via [the anonymous GitHub link](https://anonymous.4open.science/r/BSMS-GNN-ICLR-2023/). In the [ReadMe](https://anonymous.4open.science/r/BSMS-GNN-ICLR-2023/README.md), you can also find the detailed instruction on how to use the code to re-generate the RMSE we reported and the [link](https://drive.google.com/drive/folders/15UjqYdDX_Zhf-uPIs0bIZ5hYVjNiZUZy?usp=share_link) where the data and trained models are hosted.
> >
> > Thank you all, and let us know of any updates.

---

> ### Author Response · Authors · 2022-11-10
> **Detailed tables, new experiments and ablations**
>
> To dear reviewers 1, 2, and 4,
>
> Combined your suggestions on add-on experiments and detailed data, we have made the following progress:
>
> 1. Again, we sincerely thank you for requiring adding a detailed table. Here we have selected the most relevant criteria, assuming our method is to be deployed in production, and plotted them in the tables below.
> 2. We have also finished evaluating the performance of GraphUNet (Gao et al. (2019)) in all four cases (but for Font because GraphUNet is too slow).
> 3. Although Liu et al. (2021) and Fortunato et al. (2022) are impractical for full experiments as they require drawing 20,000 meshes, we abstracted 2 **transition** methods from their work and combined them into the ablation study. We have completed this ablation study on the **transition** method as well as its influence on training speed. Please see the attached tables in this thread.
>
> All the related tables and descriptions will be added in the revised version before this discussion deadline.

---

> > ### Author Response · Authors · 2022-11-15
> > **Table for detailed results and newly finished GraphUNet (Gao et al. (2019)**
> >
> > |         **Measurements**         | **Case** |     **Our's**    | **Lino et al. (2022)** | **Pfaff et al. (2020)** | **Gao et al. (2019)** |
> > |:--------------------------------:|:--------:|:----------------:|:----------------------:|:-----------------------:|:---------------------:|
> > |      Training time/step [ms]     | Cylinder |     **10.14**    |          15.36         |          19.29          |         16.20         |
> > |                                  |  Airfoil |     **18.82**    |          25.26         |          36.72          |         55.08         |
> > |                                  |   Plate  |     **15.58**    |          49.65         |          49.15          |         31.88         |
> > |                                  |    IDP   |     **45.96**    |         107.16         |          117.48         |        1,833.37       |
> > |       Infer time/step [ms]       | Cylinder |       6.75       |        **6.18**        |          14.50          |         24.30         |
> > |                                  |  Airfoil |     **8.64**     |          20.40         |          24.20          |         33.60         |
> > |                                  |   Plate  |     **14.01**    |          18.12         |          15.70          |         16.20         |
> > |                                  |    IDP   |     **33.33**    |          41.66         |          82.35          |         629.33        |
> > | Training cost [hrs], Final epoch | Cylinder |   **21.41, 19**  |        35.84, 21       |        64.30, 30        |       76.15, 39       |
> > |                                  |  Airfoil |  **122.33, 39**  |       176.82, 42       |        275.40, 45       |       206.55, 37      |
> > |                                  |   Plate  |   **56.07, 27**  |       125.78, 19       |        176.94, 27       |       127.50, 30      |
> > |                                  |    IDP   | **2.68E+01, 21** |      5.66E+01, 19      |       6.20E+01, 19      |           NA          |
> > |           RMSE-1 [1e-2]          | Cylinder |   **2.04E-01**   |        2.20E-01        |         2.26E-01        |        8.09E-01       |
> > |                                  |  Airfoil |     2.88E+01     |      **2.68E+01**      |         4.35E+01        |        2.93E+01       |
> > |                                  |   Plate  |     2.87E-02     |        2.20E-02        |       **1.98E-02**      |        2.03E-02       |
> > |                                  |    IDP   |   **1.77E-02**   |        1.87E-02        |         1.95E-02        |           NA          |
> > |          RMSE-50 [1e-2]          | Cylinder |   **2.42E+00**   |        2.74E+00        |         4.39E+00        |        1.87E+01       |
> > |                                  |  Airfoil |   **1.10E+03**   |        1.22E+03        |         1.66E+03        |        1.17E+03       |
> > |                                  |   Plate  |     3.18E-02     |      **2.78E-02**      |         2.88E-02        |        5.19E-02       |
> > |                                  |    IDP   |   **1.08E-01**   |        3.24E-01        |         1.78E-01        |           NA          |
> > |          RMSE-all [1e-2]         | Cylinder |   **8.37E+00**   |        8.49E+00        |         1.07E+01        |        1.65E+02       |
> > |                                  |  Airfoil |   **4.21E+04**   |        5.56E+04        |         6.95E+04        |        6.11E+04       |
> > |                                  |   Plate  |     1.60E-01     |      **1.48E-01**      |         1.51E-01        |        5.46E-01       |
> > |                                  |    IDP   |   **2.20E-01**   |        3.78E-01        |         3.65E-01        |           NA          |
> >
> > Caption: Detailed measurements between our method, MS-GNN-GRID, GRAPHMESHNETS, and GRAPHUNET. All measurements are conducted using a single Nvidia RTX 3090. BSMS-GNN consistently generates stable and competitive global rollouts with the smallest training cost. BSMS-GNNis also light-weighted and has the fastest inference time. It is also free from the large RMSE due to poor pooling on unseen geometries where the learnable pooling module of GRAPHUNET suffers. GRAPHUNET also can not scale up due to the adjacent matrix multiplication inside the forward process, making the training for Font more than 50hrs/epoch hence impossible for full experiments.

---

> > ### Author Response · Authors · 2022-11-17
> > **Table for ablation study on transition methods**
> >
> > | **Measurements**        |  **Ours**  |  **None**  | **Graph-Conv** | **Pos-Kernel** | **Learnable-GMP** |
> > |-------------------------|:----------:|:----------:|:--------------:|:--------------:|:-----------------:|
> > | Training time/step [ms] |    10.14   |  **9.30**  |      10.07     |      10.06     |       17.75       |
> > | Infer time/step [ms]    |    6.75    |  **5.70**  |      6.46      |      6.90      |       11.28       |
> > | Training RAM [GBs]      | **11.041** | **11.041** |   **11.041**   |   **11.041**   |       18.033      |
> > | Infer RAM [GBs]         |  **1.923** |  **1.923** |    **1.923**   |    **1.923**   |       1.931       |
> > | RMSE-1 [1e-2]           |    0.29    |  **0.15**  |      0.34      |      0.64      |        0.47       |
> > | RMSE-50 [1e-2]          |    14.30   |   205.00   |     240.00     |      17.70     |     **13.50**     |
> > | RMSE-all [1e-2]         |    16.80   |   259.00   |     551.00     |      20.10     |     **15.70**     |
> >
> > Caption: Detailed measurements of different transition methods. Our’s and Pos-Kernel(from Liu et al. (2021)) are the only two non-parametric transitions that are light-weighted and produce reliable rollouts compared to the expensive Learnable transition(from Fortunato et al. (2022)).

---

### Author Response · Authors · 2022-11-14
**Opensoursed**

We now share the code via [the anonymous GitHub link](https://anonymous.4open.science/r/BSMS-GNN-ICLR-2023/). In the [ReadMe](https://anonymous.4open.science/r/BSMS-GNN-ICLR-2023/README.md), you can also find the detailed instruction on how to use the code to re-generate the RMSE we reported and the [link](https://drive.google.com/drive/folders/15UjqYdDX_Zhf-uPIs0bIZ5hYVjNiZUZy?usp=share_link) where the data and trained models are hosted.

---

### Author Response · Authors · 2022-11-18
**Revision submitted**

Dear reviewers,
Thank you for providing valuable suggestions that helped us improve our work. We have revised our paper based on suggestions and added more experiments/ablations. We just submitted the revised version pdf, where the difference is highlighted in blue.

Below is a list of summarized updates:
- Emphasized the challenge more explicitly in the introduction (R1's, p1-2)
- Added citations to PointNet etc. (R1's, p3)
- Added an alternative view of why higher-order adjacent matrix enhancement is harmful (R1's, p3)
- Added Gao & Ji (2019) as a new full experiment, and discussed why other techniques are not practical. Detailed experiment data added (R1, R2's; p6-8, 14; Table 1,2 )
- Opensourced link description (all R's are concerned about reproductivity; p7)
- Additional ablation about the transition methods on training speed (R2, R4's; Table 3; p16)
- The proof of contact edge conservation (R1, R4's, p17)
- Improvements on typos, abbreviations, and definitions (R2, R3's)

Again, we sincerely appreciate your patience and effort in reviewing and are glad to have helpful suggestions because it helps with this work and the community.

Best,

---

### Decision · Program_Chairs · 2023-01-20

**Decision:**

Reject

**Justification For Why Not Higher Score:**

During review and rebuttal, reviewers are concerned with many aspects of this paper, including experimental comparison and results, etc. Although the authors have responded with new results, these concerns are not completely solved. Thus a reject is recommended so the authors can fully revise their paper for a future submission.

**Justification For Why Not Lower Score:**

NA

**Metareview: Summary, Strengths And Weaknesses:**

This paper proposes an improved version of graph neural nets for physical simulation. During review and rebuttal, reviewers are concerned with many aspects of this paper, including experimental comparison and results, etc. Although the authors have responded with new results, these concerns are not completely solved. Thus a reject is recommended so the authors can fully revise their paper for a future submission.